# Unveiling the mediating role of cultural trade and domestic identity in Chinese consumer engagement with foreign films and TV series

**Sheng Yuan**[1], **Wei-Lun Huang**[2]*, **Zhihan Chen**[2], **Hengbin Yin**[2]

**1** School of Humanities and Arts, Ningbo University of Technology, Ningbo, China, **2** School of Finance and Trade, Wenzhou Business College, Wenzhou, China

* huangwl@wzbc.edu.cn

**Data Availability Statement:** All relevant data are within the manuscript and its Supporting information files.

## Abstract

The research adopts a comprehensive methodological framework to investigate the influence of foreign films and TV series on the behavior of Chinese consumers. Rooted in a thorough review of existing literature, the study implements questionnaire surveys to gather primary data from 786 meticulously selected respondents. Initial data analysis through descriptive methods is subsequently refined via Partial Least Squares Structural Equation Modeling to elucidate complex interrelationships among the variables under consideration. The findings of the study indicate a growing consumer inclination towards foreign films and TV series in China. Notably, the cultural construct labeled 'Misunderstood' emerges as a critical determinant, underscoring the significance of cultural literacy in the consumption patterns of foreign films and TV series. Furthermore, the research highlights the mediating effect of domestic cultural identity on consumer preferences, which are further influenced by demographic factors such as age, gender, education, occupation, and income. By integrating economic theories of consumer choice with trade theories related to cultural exchange, the study offers an in-depth analysis of the market dynamics governing foreign films and TV series consumption in China. The implications point to substantial opportunities for content that is culturally resonant, providing valuable strategic insights for marketers and content creators operating within this complex and evolving landscape.

## Introduction

Globalization has profoundly intensified cultural exchanges, particularly through the proliferation of foreign films and TV series [1–3]. The United Nations has documented the rapid expansion of global trade in creative goods, including foreign films and TV series, which is growing at a pace that surpasses traditional merchandise trade [4, 5]. This trend is largely driven by the widespread adoption of digital technologies and the increasing availability of internet access. Against this backdrop, streaming giants like Netflix now possess a very extensive and diverse subscriber base across multiple continents. These platforms not only extend their business reach globally but also act as pivotal agents of cultural diplomacy, enhancing

**Funding:** SY was supported by the National Social Science Fund of China (http://www.nopss.gov.cn/) under Grant 19BXW102, and Ningbo University of Technology (www.nbut.edu.cn) under Grant 2023KQ087. The two sponsors did not play any role in the study design, data collection and analysis, decision to publish, or preparation of the manuscript.

**Competing interests:** The authors have declared that no competing interests exist.

global cultural literacy and promoting cross-cultural dialogues, especially in the realm of films and TV series [1, 6–8]. Thus, detailed investigations into consumer engagement with foreign films and TV series can yield vital data on viewing preferences and behaviors, providing practical advice for media marketers and content producers engaging a global audience. Moreover, China's status as the world's second largest media market [9] underscores the critical need to conduct research on its audiences to generate China-specific insights for international marketers.

In addition to industry practice, a comprehensive understanding of consumer engagement with foreign films and TV series is also essential for academic inquiry [10]. The theoretical frameworks employed in consumer behavior research offer valuable insights into the effects of engagement with foreign films and TV series. Cultural hybridization theory suggests that these media products serve as a crucible for the blending of diverse cultural norms, leading to the creation of new cultural identities. Similarly, uses and gratifications theory posits that individuals engage with these media products to satisfy specific personal needs, such as entertainment, education, and cultural enrichment. Further, theories such as the theory of planned behavior and the theory of reasoned action elucidate how attitudes, social norms, and perceived control influence individual preferences for foreign films and TV series from different cultural backgrounds. Collectively, these theories suggest that individual attitudes and interpretations of cultural values play a significant role in shaping consumption patterns [11, 12]. However, although a number of studies [13–16] have explored this role, very few have comprehensively examined the complexities of consumer engagement with foreign films and TV series, particularly considering a range of mediating and moderating variables.

Motivated by this research gap, the current study adopts a robust methodological framework, incorporating literature reviews, surveys, descriptive statistical analyses, and Partial Least Squares Structural Equation Modeling (PLS-SEM) to scrutinize consumer behavior concerning foreign films and TV series within the Chinese context. The primary objective of this research is to clarify the causal relationships between consumers' perceptions of the cultural, utilitarian, and societal values inherent in these media products and their subsequent consumption patterns. Central to this investigation is the analysis of the mediating roles played by cultural trade and domestic cultural identities in shaping these behaviors, alongside an assessment of the moderating effects exerted by individual characteristics. This approach rigorously addresses the multifaceted and dynamic nature of consumer behavior.

The study makes significant theoretical contributions across several critical domains: First, it deepens our understanding of the determinants influencing the consumption of foreign films and TV series in China. Second, it fills a crucial gap in existing literature by exploring the mediating roles of cultural trade and domestic cultural identities. Third, it adds a layer of complexity to consumer behavior research by incorporating individual attributes as moderating factors. Fourth, it offers unique insights specific to China, illuminating the distinct cultural, economic, and social dynamics within the region. Collectively, these contributions present a sophisticated, multi-dimensional framework for comprehending the consumption of foreign films and TV series in China, thereby broadening the study's relevance to various cultural contexts and enhancing both theoretical and practical knowledge.

The article is systematically organized into clearly defined sections, each dedicated to a specific aspect of the investigation into the impact of foreign films and TV series consumption on consumer behavior and cultural identity. The second section delivers an exhaustive literature review, covering seminal works on these subjects [17–22]. The third section delineates the study's theoretical framework, identifying crucial variables such as the perceived cultural, practical, and social values of foreign films and TV series, as well as consumer behavior, in

conjunction with mediating and moderating variables like cultural trade and individual characteristics.

The fourth section elaborates on the research methodology, detailing the study design, data collection techniques, and analytical methods employed. It also presents findings derived from descriptive statistics and PLS-SEM, establishing causal links between perceived values of foreign films and TV series and consumption behaviors while examining the mediating roles of cultural trade and domestic identities. The fifth section interprets the results of multigroup analysis, emphasizing the moderating effects of individual attributes. The sixth section critically assesses the findings in relation to the hypotheses, while also recognizing the study's suggestions. The final section summarizes the key findings, discusses their theoretical and practical implications, and proposes avenues for future research.

## Literature review

Foreign films and TV series have emerged as potent instruments of soft power, significantly shaping global consumer attitudes and economic landscapes. Their widespread distribution via digital streaming platforms has enabled these media to transcend national boundaries, offering diverse cultural perspectives to audiences worldwide [23–25]. Platforms such as Netflix, which reported over 223 million global subscribers in 2021, exemplify the extensive reach of foreign media content [7]. This global expansion has not only transformed cultural landscapes but also altered the ways in which cultures are perceived, integrated, and consumed on a worldwide scale [26]. The influx of foreign media content has broadened entertainment options, facilitating the exchange of cultural values and fostering cross-cultural understanding, while simultaneously challenging traditional cultural boundaries [1].

This Fig 1 presents a detailed timeline of significant cases in the development and global impact of foreign films and TV series. It highlights pivotal moments and technological advancements that have shaped the international media landscape, illustrating how these cultural products have influenced and integrated into global cinema and television. Each case in the timeline serves as a landmark in the ongoing evolution and cross-cultural exchange within the global entertainment industry. This phenomenon is rooted in Globalization Theory, Cultural Hybridization, and Technological Convergence. Globalization Theory posits that the exchange of cultural products across borders has led to a more interconnected world [27], where films like "Rashomon," "La Strada," and "Crouching Tiger, Hidden Dragon" transcend their national origins to become global icons. These films not only introduce diverse cultural narratives to international audiences but also influence global cinematic styles and trends, reinforcing the idea that cultural products serve as vehicles for cross-cultural understanding and soft power.

Cultural Hybridization suggests that as global cinema spreads, it fosters a blending of cultural elements, leading to new forms of artistic expression [28]. This is evident in international films like "The Godfather" and "Pan's Labyrinth," which incorporate and reinterpret cultural themes, resonating with global audiences while maintaining distinct cultural identities. The impact of these films is a testament to the fluidity of cultural exchange in the modern world.

Technological Convergence plays a critical role in the 21st century, particularly with the advent of AI and digital technologies in media production [29]. The creation of China's first AI-generated video series "Qianqiu Shisong" and the AI-assisted translation and adaptation of the series "Lai Long Qu Mai" in 2024 demonstrate how technology is reshaping the production and consumption of cultural products. These innovations reflect a shift towards a more digitized and automated media landscape, where AI enhances creative processes and enables broader global distribution and customization of content to meet diverse cultural preferences.

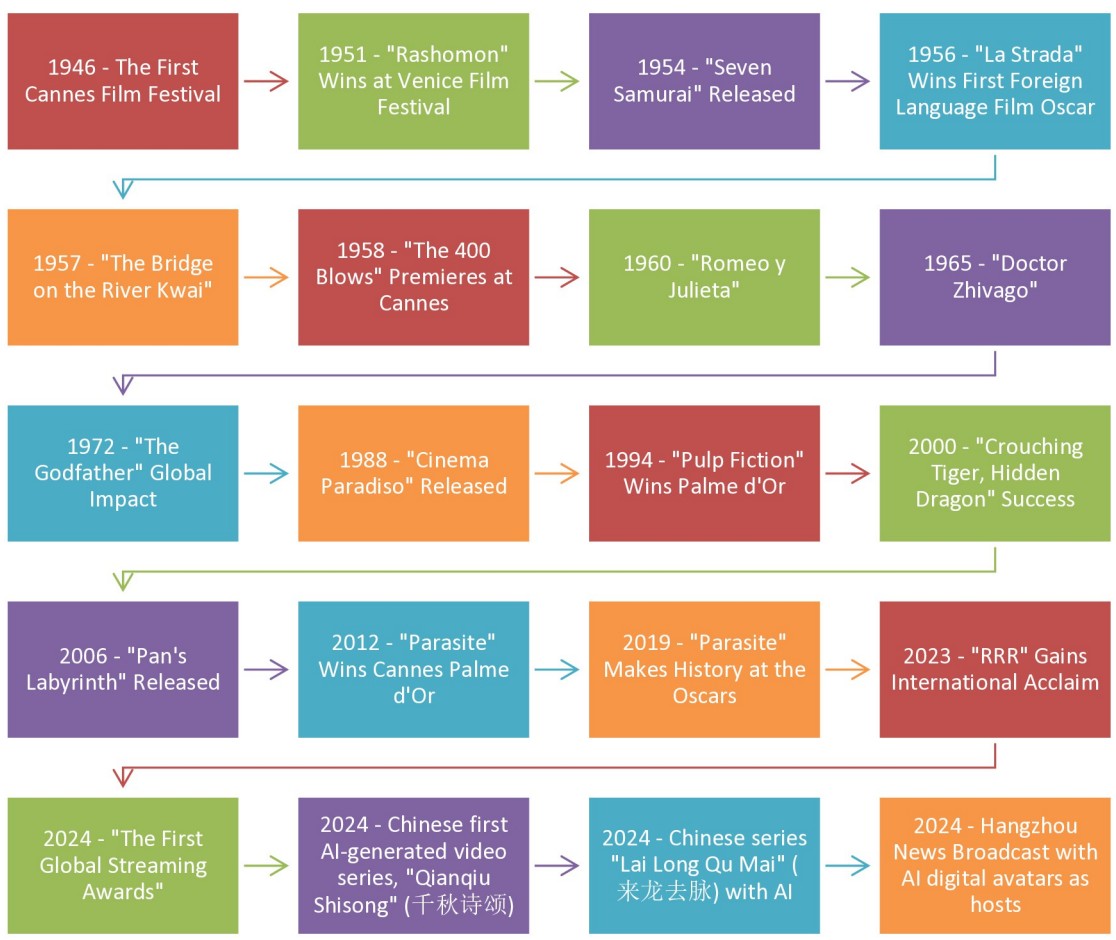

**Fig 1. Chronological timeline of key foreign films and TV series milestones.**

The introduction of AI digital avatars in news broadcasting, as seen in the Hangzhou News Broadcast [30], represents the cutting edge of media convergence. This development highlights the growing role of AI in transforming traditional media formats, making content more interactive and accessible across different platforms and cultural contexts.

The study of foreign films and TV series has a long-established history, tracing back to 1939, with initial research focusing on media's role in shaping societal norms, language evolution, and cultural identity [31, 32]. Despite these contributions, there remain significant gaps in understanding the complex relationships between cultural values, cultural trade influences, and domestic cultural identities, and how these factors collectively impact consumer behavior. Existing literature frequently overlooks the nuanced interactions between these variables, particularly their roles as mediating and moderating factors [33, 34].

Consumer behavior towards foreign films and TV series is driven by a complex network of causal relationships, where cultural, utilitarian, and social values play central roles. Cultural values significantly influence how consumers perceive and engage with foreign media. For example, Straubhaar's concept of cultural proximity suggests that audiences are more inclined to consume media that resonates with their cultural values, thereby emphasizing the importance of cultural alignment in media consumption [35]. Utilitarian values, such as the perceived quality of content, are also crucial in driving consumption decisions [36, 37]. High-

quality production, compelling narratives, and the use of advanced technology significantly enhance the perceived value of foreign content, making it more appealing to global audiences. Social values, including prestige and identity expression, further influence how consumers interact with these media products [38]. The alignment between cultural values and the narratives presented in foreign films and TV series mediates the relationship between these values and consumer behavior. Strong cultural alignment leads to greater engagement, as consumers integrate the media into their self-concept—a process supported by Social Identity Theory [39]. Conversely, cultural misalignment can lead to reduced consumption due to perceived cultural encroachment [17, 40]. These findings underscore the importance of cultural sensitivity in the production and distribution of foreign media, ensuring that content resonates with diverse global audiences.

Cultural trade serves as a pivotal mediator in the relationship between cultural values and consumer behavior. According to Cultural Trade Theory, the global dissemination of media such as foreign films and TV series enriches the entertainment industry and fosters intercultural dialogue, shaping consumer behavior by influencing perceptions of cultural and social values [41]. For instance, the global proliferation of South Korean dramas, known as Hallyu, has been instrumental in spreading Korean cultural values, impacting consumer behavior across various international markets [42]. However, this process is often moderated by domestic cultural identity. Individuals with a strong connection to their native culture may perceive foreign media as a threat, leading to diminished engagement [23]. Conversely, a positive domestic cultural identity can enhance the acceptance of foreign media, particularly when the cultural narratives align with the consumer's values, increasing engagement. In contrast, a negative domestic cultural identity can mediate this relationship by fostering resistance to foreign cultural influences, reducing consumption [40]. This duality highlights the need for media producers to consider the cultural contexts in which their content is consumed, as misalignment with domestic cultural identities can compromise the success of foreign media in local markets.

Furthermore, the relationship between cultural trade and consumer behavior is nuanced by specific consumer characteristics that influence how these interactions manifest. Age, for instance, is a critical determinant, with different age groups exhibiting unique value systems that shape their engagement with foreign media. Younger consumers often demonstrate greater openness to international content, driven by a globalized worldview and exposure to diverse cultures via digital platforms, whereas older consumers may prefer media that aligns more closely with their established cultural backgrounds [43]. Gender also plays a significant role in influencing preferences and communication styles, which can affect how different genders engage with foreign media content [43]. Education, as a moderating factor, shapes individual personality traits by fostering knowledge, competencies, and ethical values, thereby influencing media consumption patterns. Higher educational attainment is associated with enhanced critical thinking, creativity, and cultural appreciation, impacting media choices [43]. Occupation influences consumer behavior by demanding specific skills and personality traits relevant to various professional sectors, shaping preferences for certain types of media content [43]. Finally, income guide lifestyle decisions, set priorities, and shape value orientations. Higher income is linked with greater self-assurance, elevated social standing, and access to a broader range of cultural products, all of which influence media consumption behavior [43]. These consumer characteristics, as moderating variables, enrich our understanding of how different demographic segments engage with foreign films and TV series and how these interactions are shaped by the interplay between cultural trade, domestic cultural identity, and consumer behavior.

This study addresses critical gaps in the literature by systematically exploring the relationships between cultural values, cultural trade, and domestic cultural identity, particularly in how these factors influence consumer behavior. By examining the mediating role of cultural trade and domestic cultural identity, this research reveals how these factors interact to shape consumer perceptions and behaviors. Additionally, by identifying consumer characteristics such as age, gender, education, occupation, and income as essential moderating variables, the study offers a comprehensive understanding of how different demographic groups engage with foreign films and TV series. These findings are particularly relevant in the context of global media consumption, where cultural sensitivity and demographic specificity are crucial for the effective dissemination and reception of foreign media content. Consequently, this study not only refines existing theoretical models but also provides valuable insights for international marketing strategies and cultural policy development, ensuring that foreign media products resonate effectively with diverse global audiences.

## Conceptual framework and methodology

### Measurement of variables

In this study, the variables were measured meticulously to capture the intricate aspects of consumer behavior towards foreign films and TV series, ensuring that each dimension was analyzed with precision and depth. (see Table 1 and S1 Appendix).

In examining consumer interactions with foreign films and TV series, three critical metrics have been identified as essential for assessing consumer behavior, hereafter referred to as $CB_0$. The first metric is the proportional consumption of foreign films and TV series ($CB_P$), which measures the extent to which individuals engage with this content. Higher proportions suggest an increased openness to cultural diversity and a cosmopolitan outlook. The second metric focuses on the duration of engagement with foreign films and TV series ($CB_{DE}$), providing insights into the depth of consumer commitment. Individuals dedicating significant time to foreign films and TV series consumption are likely to exhibit a deeper affinity for such content, indicating a potential willingness to invest more in premium or early access. The third metric assesses the monthly financial outlay on foreign films and TV series ($CB_{AME}$), offering a perspective on both the consumer's economic status and the subjective value attributed to these entertainment experiences.

In analyzing consumer interactions with foreign films and TV series, the notion of perceived cultural values within foreign films and TV series ($CV_0$) emerges as a multifaceted construct. This construct is evaluated through several pivotal dimensions. The first dimension of cultural diversity ($CV_{CP}$) emphasizes how foreign films and TV series serve as a medium for amplifying cultural pluralism by presenting diverse cultural perspectives, lifestyles, and ethical frameworks. The second dimension of cultural awareness ($CV_{CC}$) posits that foreign films and TV series function as an educational resource, enriching viewers' understanding of the cultural norms, traditions, and practices of various nations. The third dimension of international cultural exchange ($CV_{GCI}$) highlights the role of foreign films and TV series in facilitating the mutual sharing of cultural elements and values across global borders. The fourth dimension of sociopolitical and humanistic insight ($CV_{SHC}$) clarifies that foreign films and TV series provide a window into the sociopolitical and humanistic aspects of different countries, including governance structures, economic models, and social justice issues. Lastly, the dimension of cross-cultural understanding and respect ($CV_{TUR}$) suggests that engagement with foreign films and TV series can foster a nuanced appreciation and respect for diverse cultures, reducing misunderstandings and conflicts.

**Table 1. Nomenclature of dimensions and constructs of consumer behavior and cultural perceptions in foreign films and TV series.**

| Dimension | Variable Symbol | Construct Description |
|---|---|---|
| **Consumer Behavior (CB$_0$)** | CB$_P$ | Engagement with foreign content, capturing frequency and enthusiasm for watching foreign films and TV series. |
| | CB$_{DE}$ | Depth of commitment, indicating the emotional and time-related investment in foreign content. |
| | CB$_{AME}$ | Economic and perceived value, reflecting financial expenditure and personal valuation of foreign entertainment. |
| **Cultural Value (CV$_0$)** | CV$_{CP}$ | Diversity recognition, highlighting the awareness and appreciation of varied cultures presented in media. |
| | CV$_{CC}$ | Cultural learning tool, where foreign content is seen as a medium for understanding different cultural norms. |
| | CV$_{GCI}$ | Global cultural exchange facilitator, promoting intercultural communication through content. |
| | CV$_{SHC}$ | Sociopolitical context, using content to gain insight into international sociopolitical environments. |
| | CV$_{TUR}$ | Understanding and respect, fostering empathy and respect for different cultures through foreign media. |
| **Social and Practical Values (SPV$_0$)** | SPV$_{LSCE}$ | Linguistic and cultural exchange value, assessing how content aids in cross-linguistic and cultural understanding. |
| | SPV$_{ICS}$ | Cross-cultural communication skills, recognizing the role of media in enhancing interaction across cultures. |
| | SPV$_{AD}$ | Artistic appreciation, reflecting the cultural and artistic enrichment from media consumption. |
| | SPV$_{EW}$ | Societal understanding, encouraging a broadened perspective of other societies. |
| | SPV$_{PE}$ | Personal growth and ethical development, assessing the influence on an individual's values and ethics. |
| **Cultural Trade Identities (CTI$_0$)** | CTI$_{GCI}$ | International dialogue and diversity, emphasizing the promotion of cultural diversity through global trade. |
| | CTI$_{CH}$ | Cultural preservation, underlining the importance of maintaining and promoting national cultures through trade. |
| | CTI$_{EA}$ | Economic amplification, stressing cultural trade's role in economic growth. |
| | CTI$_{ACI}$ | Cultural industry development, focusing on expanding cultural industries through trade. |
| | CTI$_{ECC}$ | Export sustainability, considering the sustainability of cultural trade practices. |
| **Domestic Cultural Identity (DCI$_0$)** | DCI$_{CU}$ | Unique cultural traits, highlighting distinctive national cultural features. |
| | DCI$_{CP}$ | Cultural pride, reflecting individuals' pride in their national culture. |
| | DCI$_{UN}$ | Cultural understanding, indicating deep personal knowledge of one's own culture. |
| | DCI$_{CK}$ | Cultural knowledge transmission, signifying the sharing and inheritance of cultural knowledge. |
| | DCI$_{CI}$ | Cultural influence, focusing on how a nation's culture impacts global awareness. |
| **Consumer Characteristics (CC$_0$)** | CC$_A$ | Age as a factor, assessing how age influences content choices. |
| | CC$_G$ | Gender impact, examining how gender affects preferences and communication regarding media consumption. |
| | CC$_E$ | Education shaping values, exploring how educational background shapes personality, values, and preferences. |
| | CC$_O$ | Occupation influence, considering how occupation-related skills and behaviors affect media choices. |
| | CC$_I$ | Income effect, reflecting how income influences lifestyle and media consumption priorities. |

This understanding of perceived cultural values aligns with some previous studies [23, 44, 45], which highlight the significant influence that audiovisual media, particularly foreign films and TV series, have on consumer perceptions of various cultures. These platforms skillfully incorporate cultural elements into their narratives and character development, in accordance with Cultivation Theory, which posits that media consumption shapes consumer attitudes and affiliations toward the cultures portrayed [46].

Beyond cultural values, the concept of perceived social and practical values within foreign films and TV series ($SPV_0$) also plays a crucial role in consumer engagement. Kim and Kim [47] identify several key dimensions within this construct. The first dimension of linguistic and skill-based cultural exchange ($SPV_{LSCE}$) suggests that foreign films and TV series act as a medium for cultural dialogue, enriching audiences by exposing them to various languages, skills, and artistic forms. The second dimension of interpersonal communication skills ($SPV_{ICS}$) posits that foreign films and TV series enhance social and communicative proficiencies, especially in cross-cultural interactions. The third dimension of artistic discernment ($SPV_{AD}$) reflects how foreign films and TV series foster a deeper understanding and appreciation of diverse artistic styles and expressions. The fourth dimension of expanded worldviews ($SPV_{EW}$) highlights how foreign films and TV series encourage critical thinking about different societies and cultures, thereby broadening intellectual perspectives. Finally, the fifth dimension of personal evolution ($SPV_{PE}$) suggests that engagement with foreign films and TV series can influence one's ethical frameworks and life philosophies, contributing to individual growth.

Nicolaou [48] asserts that digital streaming platforms act as conduits for educational enrichment, particularly concerning global cultural traits and sociocultural identities. This perspective aligns with studies by Tirasawasdichai et al. [23] and Tirasawasdichai and Pookayaporn [49], who argue that foreign films and TV series serve as key instruments for both intellectual and emotional growth. The role of foreign films and TV series in disseminating cultural and informational content is further amplified by their ability to portray global contexts through visually engaging formats.

The concept of empathy, crucial to human social interactions, also plays a significant role in the context of foreign films and TV series. Some scholars [25, 26, 50, 51] have explored how empathy enables viewers to engage vicariously with characters' emotional and cognitive experiences while maintaining their distinct perspectives. This emotional involvement enhances viewers' responses but allows them to return to their pre-existing beliefs in real-world contexts. Chen and Liu [40] applied a cognitive-emotional-attitudinal framework to explain the positive reception of Disney's "Turning Red" among Chinese audiences, noting how the film's emotional depiction of intergenerational conflicts resonated with this demographic. However, the study also highlighted the challenges of distinguishing between hybrid and indigenous cultures in such narratives.

In the discussion surrounding the impact of cultural trade, the concept of Consumer's cultural trade identities ($CTI_0$) emerges as a crucial, multifaceted framework. This schema is designed to evaluate the economic and societal implications of cultural trade through five primary dimensions. Global cultural interchange ($CTI_{GCI}$) and preservation and promotion of cultural heterogeneity ($CTI_{CH}$) highlight cultural trade's ability to catalyze international dialogue and enhance cultural pluralism, underscoring the need for mutual understanding and cooperation in forming a cohesive global community. economic amplification ($CTI_{EA}$) emphasizes the significant economic benefits, such as industry expansion and job creation, that can be achieved through proactive participation in cultural trade [41, 52, 53].

Further dimensions delve into the internally focused aspects of cultural trade. The advancement of cultural industries ($CTI_{ACI}$) dimension elucidates the crucial role that cultural trade

plays in the development of various cultural sectors, including publishing, cinematography, and the musical arts. The prosperity of these sectors contributes substantially to a nation's socioeconomic framework. Finally, the exportation of cultural commodities ($CTI_{ECC}$) dimension underscores the long-term sustainability of cultural trade, which is inherently tied to the successful exportation of cultural artifacts. In this context, a conducive infrastructure, encompassing supportive legislative frameworks and stringent intellectual property protections, becomes essential.

The concept of "Cultural Discount" serves as a critical analytical lens within global cultural trade. This notion refers to the decreased market value of foreign films and TV series products due to cultural barriers. Specific cultural elements, such as unique animation styles in foreign films and TV series, may negatively impact their market value if they lack universal appeal [1, 8]. Contrary to this perspective of cultural insularity, Alvaray [52] advocates for a more comprehensive approach to fostering national culture through the audiovisual industry, emphasizing alignment with global cultural trends and active participation in international markets. San Román [53] offers a relevant example by examining Spain's strategic use of animation as a cultural and linguistic ambassador, especially for younger demographics. The rise of Computer-Generated Imagery technology and Spain's deliberate engagement in global trade networks have propelled the nation to become the fifth-largest global supplier of animation and the second-largest in Europe. This growth signifies the emergence of a new model in transnational cinema, driven primarily by private investment and globally appealing content. This model not only ensures financial viability but also boosts export figures and earns international recognition.

The framework of domestic cultural identity ($DCI_0$) serves as a critical lens for examining an individual's deep-rooted connection to their indigenous or ancestral culture. This framework includes various cultural dimensions, such as language, traditions, historical context, and artistic expressions. A well-defined $DCI_0$ not only instills individuals with a sense of cultural pride and belonging but also fosters a sophisticated understanding of global cultural pluralism. To operationalize this construct, several key dimensions are outlined.

The dimension of cultural uniqueness ($DCI_{CU}$) emphasizes the distinct cultural attributes unique to each nation, ranging from traditional practices to modern artistic endeavors. The dimension of cultural pride ($DCI_{CP}$) encapsulates the esteem individuals derive from their cultural heritage, including both historical monuments and contemporary artistic achievements. The dimension of cultural understanding ($DCI_{UN}$) involves a comprehensive grasp of one's native culture, tracing its historical lineage and current manifestations. The dimension of cultural knowledge ($DCI_{CK}$) signifies the transgenerational transmission of cultural wisdom, thereby reinforcing a nation's collective cultural identity. Finally, the dimension of cultural influence ($DCI_{CI}$) highlights the global impact of a nation's cultural presence, from the exportation of cultural artifacts to the shaping of international sociopolitical paradigms.

Demont-Heinric [32] posits that digital streaming platforms like Netflix and Amazon Prime may unintentionally perpetuate American cultural insularity due to their limited offerings of foreign films. This raises critical questions about the role these platforms play in either promoting or inhibiting cross-cultural engagement. Feng and Luo [54] suggest that in countries with substantial cultural differences from the film's origin, movies featuring high of sexual and violent content tend to achieve higher box office returns. This underscores a complex interplay between cultural differences and the market success of foreign films. Gao [55] discusses stringent regulations imposed by China's National Radio and Television Administration aimed at curbing "deviant aesthetics" and "male effeminacy" in Chinese media and advertising. These regulations reflect the Chinese government's active role in directing the

significant sociocultural transformations affecting current norms, gender roles, and women's status in contemporary Chinese society.

In the discourse on consumer interactions with international films and television series, the framework of consumer characteristics ($CC_0$) emerges as a crucial set of moderating variables that significantly influence consumer behavior and decision-making processes. This framework includes several key dimensions: Age ($CC_A$), which functions as a critical determinant given that different age groups exhibit distinct value systems, beliefs, and priorities that inform their consumption decisions; Gender ($CC_G$), which influences preferences, communication styles, and problem-solving strategies; Education ($CC_E$), which shapes individual personality traits by fostering a diverse array of knowledge, competencies, and ethical values; Occupation ($CC_O$), which modulates consumer behavior by requiring specific skill sets and personality characteristics relevant to different vocational sectors; and Income ($CC_I$), which guides lifestyle decisions, establishes priorities, and shapes value orientations. Higher income brackets are typically associated with increased self-confidence, elevated social standing, and greater access to cultural products [43].

Empirical analysis reveals significant variations in approval rates across the dimensions of $DCI_0$, leading to its bifurcation into two categories: $DCI_{0,UPN}$ and $DCI_{0,IK}$. The former category encapsulates emotional and subjective elements like Uniqueness, Pride, and Understanding, while the latter focuses on objective aspects such as Influence and Knowledge. Essentially, $DCI_{0,UPN}$ underscores the emotional ties individuals have with their native culture, whereas $DCI_{0,IK}$ highlights the objective understanding and societal implications of domestic culture. These divergent categories may indicate differing priorities or perspectives on the importance of domestic cultural identity.

## Conceptual framework and model used for estimation

The revised conceptual framework and research design outlined in Fig 2 provide a comprehensive structure for analyzing consumer behavior toward foreign films and TV series. The model was structured to test the following paths, or hypotheses, which are integral to the research objectives:

$$CB_0 = \alpha_1 + \beta_1 CV_0 + \varepsilon_1 \tag{1}$$

$$CB_0 = \alpha_2 + \beta_2 SPV_0 + \varepsilon_2 \tag{2}$$

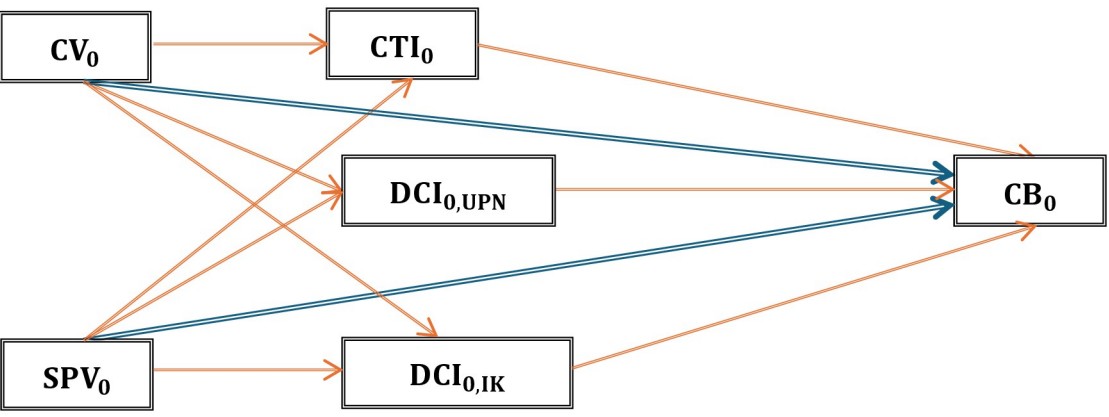

**Fig 2. The modified hypothesis framework and research design.**

$$CB_0 = \alpha_3 + \beta_3 CTI_0 + \varepsilon_3 \tag{3}$$

$$CB_0 = \alpha_4 + \beta_4 DCI_{0,UPN} + \varepsilon_4 \tag{4}$$

$$CB_0 = \alpha_5 + \beta_5 DCI_{0,IK} + \varepsilon_5 \tag{5}$$

$$CTI_0 = \alpha_6 + \beta_6 CV_0 + \beta_7 SPV_0 + \varepsilon_6 \tag{6}$$

$$DCI_{0,UNPU} = \alpha_7 + \beta_8 CV_0 + \beta_9 SPV_0 + \varepsilon_7 \tag{7}$$

$$DCI_{0,DCIDCK} = \alpha_8 + \beta_{10} CV_0 + \beta_{11} SPV_0 + \varepsilon_8 \tag{8}$$

$$CB_0 = \alpha_9 + \beta_{12} CV_0 + \beta_{13} SPV_0 + \beta_{14} CTI_0 + \beta_{15} DCI_{0,UPN} + \beta_{16} DCI_{0,IK} + \varepsilon_9 \tag{9}$$

Where $\varepsilon_1, \varepsilon_2, \varepsilon_3, \varepsilon_4, \varepsilon_5, \varepsilon_6, \varepsilon_7, \varepsilon_8, \varepsilon_9$ are residual terms.

From Fig 2, blue arrows are for $H_1$, and Eqs (1) and (2) were formulated to examine the alternative hypothesis denoted as $H_1$ ($CV_0 \rightarrow CB_0$ and $SPV_0 \rightarrow CB_0$), which posits that both $CV_0$ and $SPV_0$ exert a significant causal influence on $CB_0$. Red arrows are for $H_2$, and Eqs (3) through (9) scrutinize the alternative hypothesis labeled as $H_2$. This hypothesis asserts that $CTI_0$, as well as two subsets of $DCI_0$—namely, $DCI_{0,UPN}$ and $DCI_{0,IK}$—serve as significant mediators in the causal pathways between the independent variables ($CV_0$ and $SPV_0$) and the dependent variable ($CB_0$). Specifically, the equations test the following mediated relationships: $CV_0 \rightarrow CTI_0 \rightarrow CB_0$, $CV_0 \rightarrow DCI_{0,UPN} \rightarrow CB_0$, $CV_0 \rightarrow DCI_{0,IK} \rightarrow CB_0$, $SPV_0 \rightarrow CTI_0 \rightarrow CB_0$, $SPV_0 \rightarrow DCI_{0,UPN} \rightarrow CB_0$, and $SPV_0 \rightarrow DCI_{0,IK} \rightarrow CB_0$. Moreover, blue arrows and red arrows are for $H_3$, and they test the path coefficients' significances in multi-group analysis of $CC_0$ ($CC_A$, $CC_G$, $CC_E$, $CC_O$, and $CC_I$).

The model includes a series of equations that represent the direct and indirect relationships among various variables. The first set of equations assesses the direct impact of perceived cultural values and social and practical values on consumer behavior. These variables are critical in understanding how consumers perceive and interact with foreign films and TV series.

The second set of equations introduces the mediating roles of consumer's cultural trade identities and domestic cultural identity, which are further divided into two dimensions: "uniqueness, pride, and understanding" and "influence and knowledge". These dimensions are hypothesized to mediate the relationships between cultural values, social and practical values, and consumer behavior. The equations test the pathways through which consumer's cultural trade identities and domestic cultural identity influence consumer behavior, providing insights into the underlying mechanisms of cultural influence.

The final set of equations explores the moderating effects of various consumer characteristics, such as age, gender, education, occupation, and income, on these relationships. These characteristics are crucial for understanding the variability in consumer behavior across different demographic groups.

## Research design and data collection procedures

The study aims to clarify the causal relationships between consumers' perceived cultural, practical, and social values of foreign films and TV series and their subsequent behaviors. It also

examines the mediating roles of $CTI_0$ and $DCI_0$, as well as the moderating effects of consumer characteristics.

PLS-SEM was operationalized using specialized software such as SmartPLS, which provides advanced tools for model estimation and hypothesis testing. This approach is particularly advantageous for consumer behavior studies, as it allows for the handling of complex models and provides precise estimates even with limited sample sizes. The incorporation of Multi-group Analysis (MGA) enables the identification of group-specific variations, offering a more nuanced understanding of the factors that influence consumer behavior in the foreign films and TV series sector [12, 56–60].

To analyze the collected data, a combination of descriptive statistics, PLS-SEM, and MGA was employed. PLS-SEM is particularly well-suited for this type of research as it allows for the examination of complex models involving multiple variables. This method is widely used in social science research for its ability to handle latent variables and assess the relationships between them. The use of MGA with PLS-SEM adds an additional layer of analysis, enabling the examination of differences in consumer behavior across different demographic groups.

To effectively apply PLS-SEM in this study, the process begins with a detailed establishment of the conceptual framework and the estimation model. (see Fig 3) This involves constructing a structural model that captures the hypothesized causal relationships among the key variables under investigation. These variables, which include consumers' perceived cultural value, perceived social and practical value, cultural trade identity, and domestic cultural identity, were hypothesized to exert various influences on consumer behavior through distinct pathways.

The next step involves designing a precise measurement model for each latent variable. This model specifies how each latent construct is represented by its respective indicators or observed variables. For instance, the perceived cultural value may be assessed through dimensions such as cultural diversity, cultural awareness, and international cultural exchange, ensuring that each dimension is meticulously measured and represented.

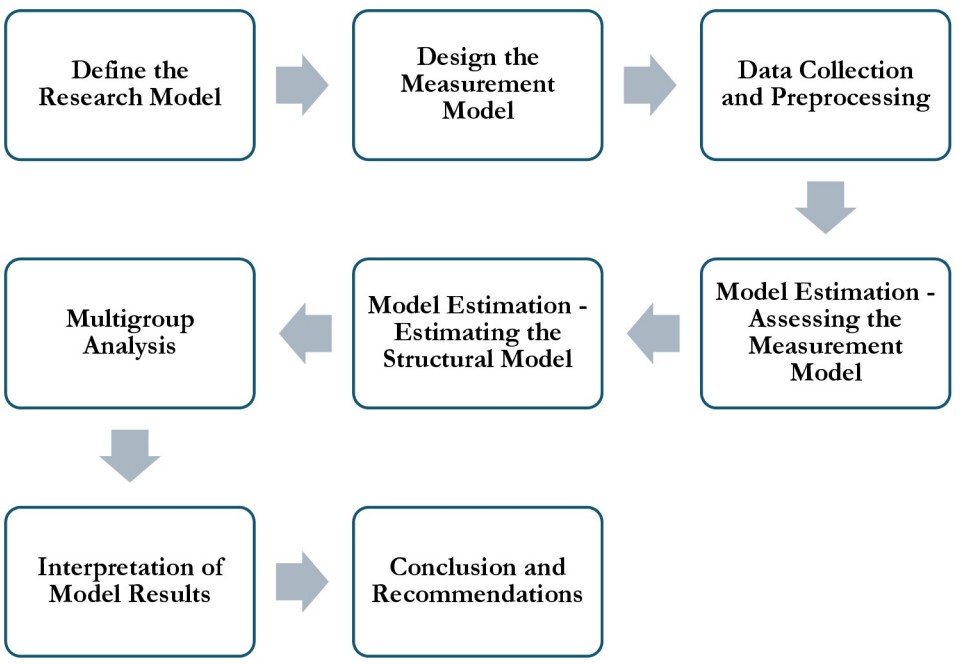

**Fig 3. The detailed diagram of the PLS-SEM and conceptual framework.**

Data collection and preprocessing are critical components of the process. Data was gathered in March 2024 through an online survey carefully crafted to accurately capture respondents' perceived values, $CTI_0$ and $DCI_0$, as well as their behavioral tendencies towards foreign films and TV series. Before conducting the survey, we obtained a waiver for ethical approval from the Academic Committee of Ningbo University of Technology, who believed that none of the survey questions would pose any psychological or physiological risks to respondents.

In this study, a non-probability sampling technique, specifically convenience sampling, was employed to select respondents who were habitual Chinese viewers of foreign films and TV series. Convenience sampling is a widely-used method when the target population is easily accessible and the research requires quick data collection. The respondents were selected based on their availability and willingness to participate in the study, ensuring that they represented a diverse demographic in terms of age, gender, education, and occupation. This diversity was essential to capture a broad spectrum of consumer behavior and preferences concerning foreign films and TV series in the Chinese context. The sample aimed to include individuals with varying exposure to international media, ensuring that the study results would be comprehensive and applicable across different segments of the population.

The minimum sample size for this study was determined using guidelines from SEM literature, particularly for PLS-SEM. According to Hair et al. [61], the minimum sample size in PLS-SEM should be ten times the number of indicators of the most complex construct in the model or the largest number of structural paths directed at a particular construct in the model. For this study, considering the complexity of the model and the number of indicators used to measure constructs like perceived cultural value, perceived social and practical value, cultural trade identity, and domestic cultural identity, a minimum sample size of 200 was deemed necessary. However, to increase the reliability and validity of the findings, the actual sample size used was 786, well above the minimum threshold, providing a robust dataset for analysis (S1 Table).

Once collected, the data underwent thorough cleaning and preparation to ensure it is suitable for analysis within the PLS-SEM framework. The estimation process in PLS-SEM was executed using specialized software, such as SmartPLS, which facilitated both the assessment of the measurement model and the estimation of the structural model. The measurement model was evaluated for reliability and validity through indicators like outer loadings, composite reliability, and average variance extracted. The structural model was then estimated by calculating path coefficients using the partial least squares method, followed by an evaluation of the model's goodness of fit.

MGA plays a significant role in this methodology, allowing for the examination of differences in path coefficients across various demographic groups, such as different age groups, genders, or educational levels. This analysis provides deeper insights into how consumer behavior may vary across different segments, enriching the interpretation of the results.

The interpretation phase involves a careful examination of key path coefficients and a discussion of their statistical significance and practical implications. For example, the study might reveal that perceived cultural value has a significant positive impact on consumer behavior, while perceived social and practical value may have a less pronounced effect.

## Results

The demographic profile of the respondents was analyzed through descriptive statistics, yielding the following insights: the average age ($CC_A$) was 31.6 years, with 31.8% of respondents aged 35 or older ($CC_{A,E}$) and 68.2% aged 34 or younger ($CC_{A,Y}$). Gender ($CC_G$) distribution was fairly balanced, with 48.9% male ($CC_{G,M}$) and 51.1% female ($CC_{G,F}$) respondents. In terms

**Table 2. The analytical overview of the variables pertinent to consumer behavior in the context of foreign films and TV series.**

| Variables | | Measurement | Approval | Mean (S.E.) | OL | VIF |
|---|---|---|---|---|---|---|
| $CB_0$ | | $CB_P$ | - | 22.73% (25.00%) | 0.91 | 2.48 |
| | | $CB_{DE}$ | - | 2.34 (1.46) hours | 0.87 | 2.14 |
| | | $CB_{AME}$ | - | RMB 55.22 (108.66) | 0.78 | 1.55 |
| $CV_0$ | | $CV_{CP}$ | 31.7% | 2.85 (1.37) | 0.89 | 3.21 |
| | | $CV_{CC}$ | 40.5% | 2.86 (1.47) | 0.84 | 2.36 |
| | | $CV_{GCI}$ | 27.4% | 2.59 (1.46) | 0.89 | 3.23 |
| | | $CV_{SHC}$ | 31.4% | 2.58 (1.41) | 0.91 | 3.51 |
| | | $CV_{TUR}$ | 29.4% | 2.71 (1.29) | 0.79 | 1.88 |
| $SPV_0$ | | $SPV_{LSCE}$ | 29.5% | 2.51 (1.51) | 0.93 | 4.76 |
| | | $SPV_{ICS}$ | 29.6% | 2.61 (1.41) | 0.92 | 4.32 |
| | | $SPV_{AD}$ | 36.1% | 3.09 (1.33) | 0.81 | 2.17 |
| | | $SPV_{EW}$ | 34.4% | 2.84 (1.31) | 0.89 | 3.13 |
| | | $SPV_{PE}$ | 31.0% | 2.63 (1.45) | 0.88 | 3.14 |
| $CTI_0$ | | $CTI_{GCI}$ | 31.9% | 2.94 (1.24) | 0.81 | 2.13 |
| | | $CTI_{CH}$ | 34.0% | 2.93 (1.31) | 0.87 | 2.90 |
| | | $CTI_{EA}$ | 35.8% | 2.92 (1.24) | 0.83 | 2.73 |
| | | $CTI_{ACI}$ | 22.6% | 2.47 (1.33) | 0.85 | 2.25 |
| | | $CTI_{ECC}$ | 34.2% | 3.26 (0.83) | 0.79 | 2.13 |
| $DCI_0$ | $DCI_{0,UPN}$ | $DCI_{CU}$ | 77.4% | 4.11 (0.98) | 0.70 | 1.17 |
| | | $DCI_{CP}$ | 79.5% | 4.28 (0.82) | 0.86 | 1.80 |
| | | $DCI_{UN}$ | 79.6% | 4.17 (0.84) | 0.83 | 2.17 |
| | $DCI_{0,IK}$ | $DCI_{CK}$ | 17.4% | 2.24 (1.19) | 0.92 | 1.87 |
| | | $DCI_{CI}$ | 18.6% | 2.25 (1.16) | 0.91 | 1.87 |

of educational attainment ($CC_E$), 40.3% of respondents held a college degree or higher ($CC_{E, M}$), while the remaining 59.7% had lower education ($CC_{E,L}$). Occupation status ($CC_O$) showed that 38.4% of respondents were in full-time employment ($CC_{O,F}$), with the rest engaged in various other occupations ($CC_{O,O}$). The average annual income ($CC_I$) was RMB 342,239.2, with 34.5% of respondents earning RMB 300,000 or more ($CC_{I,H}$), and 65.5% earning below that threshold ($CC_{I,L}$).

Consumer engagement with foreign films and TV series ($CB_0$) was assessed through three behavioral metrics. The first metric, the proportion of engagement ($CB_P$), had a mean value of 22.73% with a standard deviation of 25.00%. The second metric, average viewing time ($CB_{DE}$), was 2.34 hours, with a standard deviation of 1.46 hours. The third metric, financial expenditure ($CB_{AME}$), averaged RMB 55.22, with a standard deviation of RMB 108.66. Notably, 64.12% of respondents did not spend money on foreign films and TV series, while the rest averaged a monthly expenditure of RMB 141.37. These findings reflect a growing interest in foreign films and TV series, which holds significant economic potential for the industry, with rising popularity potentially driving sectoral growth.

From Table 2, analysis of consumer recognition of cultural values within foreign films and TV series ($CV_0$) revealed that the most highly recognized cultural value was "Cultural Awareness ($CV_{CC}$)," with 40.5% of respondents identifying this as important. Other significant values included cultural diversity ($CV_{CP}$, 31.7%), sociopolitical and humanistic insight ($CV_{SHC}$, 31.4%), cross-cultural understanding ($CV_{TUR}$, 29.4%), and international cultural exchange ($CV_{GCI}$, 27.4%). The most recognized theme among consumers was "Misunderstood," with a 57.4% recognition rate. This emphasizes the importance of cultural values in the consumption

of foreign films and TV series and highlights the growing consumer preference for content that promotes cultural diversity and awareness. The relatively high average recognition proportions suggest that cultural values are central to consumer decision-making, providing a strong incentive for producers and distributors to incorporate such themes into their content to appeal to a wider audience and enhance their market presence.

The analysis of consumer recognition in Table 2 underscores the significant role that perceived social and practical values ($SPV_0$) play in driving engagement with foreign films and TV series. The findings show that artistic quality and aesthetic appeal ($SPV_{AD}$) are highly valued by consumers, as evidenced by the highest recognition for the artistic discernment variable at 36.1%. This indicates that consumers are not only drawn to the visual and creative aspects of foreign media but also appreciate its potential to broaden their horizons by exposing them to new cultures, ideas, and perspectives. This broadening effect is particularly relevant to consumers' desire for personal development, communication skills, and foreign language proficiency, which are tied to their consumption habits.

Similarly, consumer recognition of cultural trade identities reveals an emphasis on the economic benefits of cultural exchange. Variables such as economic amplification ($CTI_{EA}$) and exportation of cultural commodities ($CTI_{ECC}$) received high recognition, highlighting the consumer perception that cultural trade fosters job creation, economic growth, and national competitiveness. Additionally, the promotion of cultural diversity and tolerance through the cross-border dissemination of cultural products is highly regarded, indicating that consumers see cultural trade as a strategic tool for strengthening national identity and contributing to global economic dynamics.

In terms of domestic cultural identities ($DCI_0$), the data shows a strong identification with and pride in one's native culture, as seen in the high recognition proportions for cultural uniqueness ($DCI_{CU}$), pride ($DCI_{CP}$), and understanding ($DCI_{UN}$), each exceeding 77%. This suggests that consumers place considerable value on preserving and respecting their domestic cultural heritage. However, the lower recognition for cultural influence ($DCI_{CI}$) and knowledge ($DCI_{CK}$) point to a potential gap in the broader understanding and engagement with the historical and cultural significance of their native heritage. This disparity may suggest that while consumers feel a personal connection to their domestic culture, there may be a lack of deeper awareness regarding its broader societal impact.

The research instruments used in this study demonstrate high reliability and validity, as evidenced by the outer loadings (OL), which all exceed the 0.6 benchmark. The variance inflation factors (VIFs) for all variables are below the critical threshold of 10, indicating that multicollinearity is not a concern. These metrics confirm the robustness of the measurements, ensuring the reliability of the statistical analyses performed.

Table 3 reinforces the reliability and convergent validity of the variables. The Cronbach's Alpha ($\alpha$) values surpass the 0.7 threshold, indicating strong internal consistency, which is further supported by Dillon-Goldstein's rho (rho_A) values. The composite reliability (CR)

**Table 3. The reliability and convergent validity metrics associated with the variables that govern consumer behavior in the realm of foreign films and TV series.**

| Variables | | $\alpha$ | rho_A | CR | AVE | $R^2$ | Adj. $R^2$ |
|---|---|---|---|---|---|---|---|
| $CB_0$ | | 0.82 | 0.84 | 0.89 | 0.74 | 0.76 | 0.76 |
| $CV_0$ | | 0.91 | 0.92 | 0.94 | 0.74 | - | - |
| $SPV_0$ | | 0.93 | 0.94 | 0.95 | 0.79 | - | - |
| $CTI_0$ | | 0.89 | 0.89 | 0.92 | 0.69 | 0.74 | 0.74 |
| $DCI_0$ | $DCI_{0,UPN}$ | 0.71 | 0.72 | 0.84 | 0.64 | 0.50 | 0.50 |
| | $DCI_{0,IK}$ | 0.81 | 0.81 | 0.91 | 0.84 | 0.44 | 0.44 |

metrics for all variables also exceed the 0.7 benchmark, while the average variance extracted (AVE) values surpass the 0.5 criterion, confirming the methodological rigor of the study. These results enhance the credibility and robustness of the findings.

The PLS-SEM analysis provides additional insights into Chinese consumer behavior in relation to foreign films and TV series (see Fig 4). Hypothesis 1 ($H_1$) receives partial support, with a strong positive relationship observed between perceived cultural values and consumer behavior towards foreign films and TV series ($CV_0 \rightarrow CB_0$), with a path coefficient of 0.60. This suggests that cultural consumption allows consumers to express their identities and values, offering them a platform to explore and integrate diverse cultural paradigms. However, the relationship between perceived practical and social values and consumer behavior ($CV_0 \rightarrow CB_0$) was found to be statistically insignificant. This could indicate that while cultural values

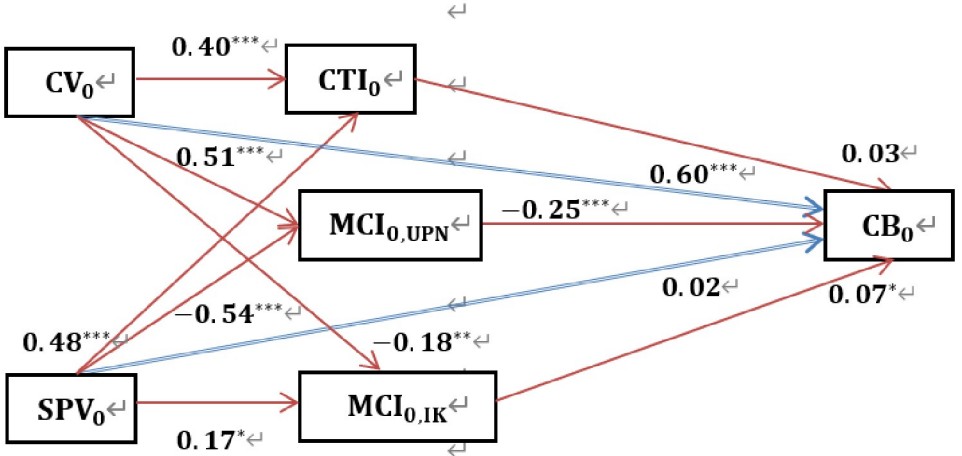

| | mean |
|---|---|
| $CV_0 \rightarrow CTI_0 \rightarrow CB_0$ | 0.01 |
| $CV_0 \rightarrow DCI_{0,UPN} \rightarrow CB_0$ | 0.04* |
| $CV_0 \rightarrow DCI_{0,IK} \rightarrow CB_0$ | 0.03* |
| $SPV_0 \rightarrow CTI_0 \rightarrow CB_0$ | 0.02 |
| $SPV_0 \rightarrow DCI_{0,UPN} \rightarrow CB_0$ | 0.13*** |
| $SPV_0 \rightarrow DCI_{0,IK} \rightarrow CB_0$ | 0.01 |

**Fig 4. The results of PLS-SEM analysis.** Note: *** p < 1%, ** p < 5%, * p < 10%.

are closely tied to personal identity and provide a meaningful emotional experience, practical and social values may not have the same influence on consumer behavior, as they do not offer the same depth of personal enrichment or emotional engagement.

Hypothesis 2 ($H_2$) is substantiated to a degree, as evidenced by the statistically significant positive mediating effects of the emotional and subjective dimensions of domestic cultural identity on both cultural values and practical and social values in shaping consumer behavior towards foreign films and TV series. Specifically, the indirect effects in the pathways from cultural values to domestic cultural identity to consumer behavior, and from practical and social values to domestic cultural identity to consumer behavior ($CV_0 \rightarrow DCI_{0,UPN} \rightarrow CB_0$ and $SPV_0 \rightarrow DCI_{0,UPN} \rightarrow CB_0$), are measured at 0.04 and 0.13, respectively. Furthermore, a notable positive indirect effect is observed in the pathway from cultural values to consumer behavior through the knowledge-based dimension of domestic cultural identity ($CV_0 \rightarrow DCI_{0,IK} \rightarrow CB_0$), quantified at 0.03.

In contrast, the mediation effects involving cultural trade identity do not demonstrate significant indirect effects. The pathways from cultural values and practical and social values to consumer behavior via cultural trade identity, and from practical and social values to consumer behavior through knowledge-based domestic identity, do not yield statistically significant results. This indicates that consumers tend to align their engagement with foreign films and TV series more closely with their domestic cultural identity rather than cultural trade identity. Cultural trade identity, as a mediator, does not appear to have a direct impact on consumer behavior. This might suggest that the influence of cultural trade identity is secondary to perceived cultural values or that its impact is mediated by additional variables not captured in the current study.

The analysis reveals strong positive relationships between cultural values and cultural trade identity, cultural values and domestic cultural identity, practical and social values and cultural trade identity, and practical and social values and knowledge-based domestic identity ($CV_0 \rightarrow CTI_0$, $CV_0 \rightarrow DCI_{0,UPN}$, $SPV_0 \rightarrow CTI_0$, and $SPV_0 \rightarrow DCI_{0,IK}$). These relationships are reflected in path coefficients of 0.40, 0.51, 0.48, and 0.17, respectively. On the other hand, negative path coefficients are noted between cultural values and knowledge-based domestic identity, and between practical and social values and emotional domestic identity ($CV_0 \rightarrow DCI_{0,IK}$ and $SPV_0 \rightarrow DCI_{0,UPN}$), with coefficients of -0.18 and -0.54, respectively. These findings underscore that consumer engagement in cultural trade related to foreign films and TV series is shaped primarily by their cultural, practical, and social values. When consumers perceive foreign films and TV series as reflecting values congruent with their own cultural ideals—such as traditional reverence or national pride—they are more likely to participate in cultural trade activities associated with these media products. Likewise, the perceived practical and social benefits of foreign films and TV series, including their entertainment value or alignment with contemporary trends, drive consumer involvement in cultural trade.

Cultural capital, encompassing knowledge, skills, and cultural experiences, provides both social and economic benefits. While engagement with foreign films and TV series can enhance this capital and strengthen cultural identity, it also introduces complexities within the framework of cultural globalization. The study concludes that consumer preferences for foreign films and TV series are largely driven by cultural congruence rather than by utilitarian or social considerations. Emotional ties to domestic cultural identity play a crucial mediating role between perceived cultural values and consumer behavior.

The values embodied in foreign films and TV series significantly impact cultural trade identities; however, the reverse does not hold, indicating a unidirectional relationship. Cultural values exert a profound influence on consumer identities, emotions, and behaviors regarding foreign films and TV series. Although exposure to foreign media may reduce the perceived

value of domestic cultural identity in some contexts, it does not eliminate an individual's overall appreciation for their native culture in other areas of life. These findings contribute to a deeper understanding of the intricate dynamics between cultural trade, domestic cultural identity, and consumer engagement with foreign films and TV series.

## Multi-group analysis

Age is a key determinant of consumer behavior, as individuals across different age groups display divergent value systems, beliefs, and priorities. These differences substantially affect their acceptance of foreign cultural content. Younger consumers are generally more open to diverse cultural influences, largely due to their exposure to globalization and their access to foreign cultures through digital platforms. Conversely, older consumers tend to favor content that aligns closely with their established cultural backgrounds. Hence, age serves as a significant factor in determining both the acceptance of and engagement with foreign films and television. The MGA in this study effectively highlights the pronounced distinctions across different age groups.

The current study conducts a multi-group analysis to investigate the moderating effects of demographic factors, particularly age, on consumer behavior towards foreign films and TV series. Employing both two-way ANOVA and PLS-SEM, the analysis reveals significant age-related moderating effects on several key behavioral pathways ($CB_{DE}$, $CV_0 \rightarrow CB_0$, $SPV_0 \rightarrow DCI_{0,UPN} \rightarrow CB_0$, $CV_0 \rightarrow CTI_0 \rightarrow CB_0$, and $SPV_0 \rightarrow CTI_0 \rightarrow CB_0$).

As highlighted in Table 4, age is found to significantly moderate relationships such as duration of engagement with foreign films and TV series, cultural values' influence on consumer behavior, and the mediating effects of domestic cultural identity and cultural trade identity on consumer engagement. Therefore, Hypothesis 3a ($H_{3a}$) receives partial empirical support, demonstrating that age plays a critical role in moderating the complex dynamics between cultural trade, domestic identity, and consumer behavior towards foreign films and TV series.

**Table 4. Multi-group analysis of respondent's age.**

| | $CC_{A,E}$ | $CC_{A,Y}$ | Difference |
|---|---|---|---|
| | **Mean** | | |
| $CB_P$ | 21.75% | 23.18% | 1.43% |
| $CB_{DE}$ | 1.89 hour | 2.56 hour | -0.67*** |
| $CB_{AME}$ | RMB 55.40 | RMB 55.13 | RMB 0.27 |
| | Path Coefficient | | |
| $CV_0 \rightarrow CB_0$ | 0.34*** | 0.62*** | -0.27* |
| $SPV_0 \rightarrow CB_0$ | 0.14* | 0.02 | 0.12 |
| | Specific indirect effects | | |
| $SPV_0 \rightarrow DCI_{0,UPN} \rightarrow CB_0$ | 0.05 | 0.15*** | -0.10* |
| $CV_0 \rightarrow DCI_{0,IK} \rightarrow CB_0$ | 0.08 | 0.01 | 0.07 |
| $CV_0 \rightarrow CTI_0 \rightarrow CB_0$ | 0.11* | -0.01 | 0.12* |
| $SPV_0 \rightarrow DCI_{0,IK} \rightarrow CB_0$ | 0.04 | 0.01 | 0.04 |
| $CV_0 \rightarrow DCI_{0,UPN} \rightarrow CB_0$ | 0.11** | 0.04 | 0.07 |
| $SPV_0 \rightarrow CTI_0 \rightarrow CB_0$ | 0.08* | -0.01 | 0.09* |

Note:

*** p < 1%,

** p < 5%,

* p < 10%.

This finding introduces additional complexity to existing research, emphasizing the importance of age-specific strategies in addressing variations in consumer engagement with foreign films and TV series.

The moderating effects of age on consumer behavior is attributed to evolving life-stage factors. Consumer preferences and behaviors adapt with changes in financial circumstances, cultural exposure, and personal priorities. Younger consumers, who are in the process of shaping their cultural identities, tend to be more receptive to foreign films and TV series that align with global and culturally diverse themes. They are more likely to invest time in consuming culturally relevant content and respond to targeted marketing that appeals to their evolving sense of identity. In contrast, older consumers often focus on more practical considerations, becoming less engaged with foreign films and TV series that do not resonate with their established values. Their behavior reflects a more utilitarian approach, where the emotional connections typically formed through cultural engagement play a lesser role. These generational differences underline the need for distinct marketing strategies and content curation that cater to the specific needs and values of different age groups.

The divergent consumption patterns between younger and older consumers are shaped by distinct cultural values and personal experiences. Younger generations, growing up in an increasingly globalized environment, are more inclined to embrace cultural diversity. Their consumption of foreign films and TV series is heavily influenced by emotional connections to both global and domestic cultures. In contrast, older consumers, with a lifetime of experiences fostering pragmatism, tend to prioritize practical and social values over emotional connections. Their engagement with foreign films and TV series is often evaluated more objectively, with less emphasis on cultural resonance. These age-based variations in consumer behavior emphasize the necessity of tailored marketing and content strategies that reflect the distinct preferences of younger and older audiences.

Gender influences individual preferences and communication styles, thereby affecting engagement with foreign audiovisual content. For instance, men and women may exhibit distinct preferences regarding media genres, with men often favoring action or sports-related content, while women may prefer drama or romance. Additionally, cultural roles and societal expectations linked to gender may moderate the degree of acceptance of foreign content. Thus, gender functions as a moderating variable that aids in comprehensively understanding the differences in the consumption of foreign cultural products.

The moderating effects of gender on consumer behavior towards foreign films and TV series, as shown in Table 5, reveal that gender significantly moderates several aspects of consumer behavior ($CB_P$, $CB_{DE}$, $CB_{AME}$, $SPV_0 \rightarrow DCI_{0,UPN} \rightarrow CB_0$, $CV_0 \rightarrow DCI_{0,IK} \rightarrow CB_0$, $CV_0 \rightarrow CTI_0 \rightarrow CB_0$, $CV_0 \rightarrow DCI_{0,UPN} \rightarrow CB_0$, and $SPV_0 \rightarrow CTI_0 \rightarrow CB_0$), including engagement proportions, duration of engagement, and financial expenditure. Gender also affects the mediating roles of perceived social and practical values on domestic cultural identity and consumer behavior, as well as the impact of perceived cultural values and cultural trade identity on consumer behavior.

This analysis offers partial validation of Hypothesis 3b (**H$_{3b}$**), emphasizing the significant role of gender in shaping consumer behavior towards foreign media. Gender differences affect not only the types of content consumed but also the underlying motivations and values that drive such consumption. Therefore, recognizing these gender-specific moderating effects is essential for both theoretical research and practical applications in the media industry.

Cultural subtleties and entrenched gender stereotypes further shape these consumption patterns. For instance, foreign media content that highlights masculine values may have a stronger appeal to male audiences and a lesser impact on female viewers. Traditional gender norms also influence content preferences; males may favor action-oriented or sports-related foreign

**Table 5. Multi-group analysis of respondent's gender.**

| | $CC_{G,M}$ | $CC_{G,F}$ | Difference |
|---|---|---|---|
| | **Mean** | | |
| $CB_P$ | 28.26% | 17.44% | 10.81%*** |
| $CB_{DE}$ | 2.54 hour | 2.16 hour | 0.38*** |
| $CB_{AME}$ | RMB 61.85 | RMB 48.88 | 12.97*** |
| | Path Coefficient | | |
| $CV_0 \rightarrow CB_0$ | 0.51*** | 0.69*** | -0.17 |
| $SPV_0 \rightarrow CB_0$ | 0.05 | -0.01 | 0.06 |
| | Specific indirect effects | | |
| $SPV_0 \rightarrow DCI_{0,UPN} \rightarrow CB_0$ | 0.17*** | 0.05 | 0.12** |
| $CV_0 \rightarrow DCI_{0,IK} \rightarrow CB_0$ | -0.01 | 0.08* | -0.09** |
| $CV_0 \rightarrow CTI_0 \rightarrow CB_0$ | 0.10*** | -0.11*** | 0.21*** |
| $SPV_0 \rightarrow DCI_{0,IK} \rightarrow CB_0$ | -0.01 | -0.02 | 0.00 |
| $CV_0 \rightarrow DCI_{0,UPN} \rightarrow CB_0$ | -0.04* | 0.20*** | -0.24*** |
| $SPV_0 \rightarrow CTI_0 \rightarrow CB_0$ | 0.15*** | -0.08** | 0.24*** |

Note:

*** p < 1%,

** p < 5%,

* p < 10%.

media, perceiving these genres as status symbols, while females might prefer drama or romance, thereby allocating their resources differently. Additionally, gender affects how cultural values in foreign media are interpreted; males might be more receptive to dominant cultural narratives, whereas females might engage with these narratives in a more critical manner. This nuanced understanding is pivotal for advancing both academic research and strategic planning in the media sector.

Educational attainment significantly impacts an individual's value system, cultural cognition, and media consumption behavior. Higher education are generally correlated with greater critical thinking abilities and cultural appreciation, enabling individuals to understand foreign cultural content more profoundly, including its social and economic dimensions. Highly educated consumers tend to show a greater acceptance and appreciation for foreign films and television due to their capacity to derive cultural and intellectual value from such content.

The examination of educational background's impact on foreign films and TV series consumption patterns (see Table 6) illustrates that educational attainment significantly moderates various facets of consumer behavior ($CB_P$, $CB_{DE}$, $CB_{AME}$, $CV_0 \rightarrow CB_0$, $SPV_0 \rightarrow CB_0$, $CV_0 \rightarrow DCI_{0,IK} \rightarrow CB_0$, $CV_0 \rightarrow CTI_0 \rightarrow CB_0$, $SPV_0 \rightarrow DCI_{0,IK} \rightarrow CB_0$, and $SPV_0 \rightarrow CTI_0 \rightarrow CB_0$), including the proportion of engagement, duration of engagement, and average monthly expenditure. Additionally, educational background affects the relationships between perceived cultural values and consumer behavior, as well as the mediating effects of domestic cultural identity and cultural trade identity.

The findings provide partial support for Hypothesis 3c (**H3c**), emphasizing the substantial role of educational background in shaping consumer engagement with foreign media. Educational attainment influences not only direct consumption behaviors but also the complex interplay between cultural and practical values, domestic identity, and cultural trade identity. This nuanced understanding underscores the importance of considering education in future research and practical applications within the media and entertainment sectors.

**Table 6. Multi-group analysis of respondent's education.**

| | $C_{C,E,M}$ | $C_{C,E,L}$ | Difference |
|---|---|---|---|
| | **Mean** | | |
| $CB_P$ | 29.34% | 18.26% | 11.08%*** |
| $CB_{DE}$ | 2.58 hour | 2.18 hour | 0.40*** |
| $CB_{AME}$ | RMB 66.25 | RMB 47.76 | 18.48* |
| | **Path Coefficient** | | |
| $CV_0 \rightarrow CB_0$ | 0.78*** | 0.42*** | 0.36*** |
| $SPV_0 \rightarrow CB_0$ | -0.09* | 0.11 | -0.20* |
| | **Specific indirect effects** | | |
| $SPV_0 \rightarrow DCI_{0,UPN} \rightarrow CB_0$ | 0.12*** | 0.13*** | -0.01 |
| $CV_0 \rightarrow DCI_{0,IK} \rightarrow CB_0$ | 0.07 | -0.02* | 0.09** |
| $CV_0 \rightarrow CTI_0 \rightarrow CB_0$ | -0.10*** | 0.09** | -0.20*** |
| $SPV_0 \rightarrow DCI_{0,IK} \rightarrow CB_0$ | 0.09 | 0.00** | 0.09*** |
| $CV_0 \rightarrow DCI_{0,UPN} \rightarrow CB_0$ | 0.00* | 0.04 | -0.04 |
| $SPV_0 \rightarrow CTI_0 \rightarrow CB_0$ | -0.05*** | 0.14** | -0.19*** |

Note:

*** $p < 1\%$,

** $p < 5\%$,

* $p < 10\%$.

Consumers with higher education and incomes demonstrate increased demand for foreign films and TV series, reflecting their refined appreciation for high culture and diverse international content. Their advanced education enables a critical assessment of media content and consideration of its social and economic implications. These consumers prioritize the practical and social benefits of media consumption. Conversely, individuals with lower educational attainment may lack the necessary insight to connect perceived values with consumption behaviors effectively. This discrepancy highlights the significant influence of educational background on media consumption patterns and its impact on how perceived values, domestic cultural identity, and cultural trade identity shape consumer behavior.

Occupation determines not only an individual's economic capacity but also shapes personality traits and lifestyle preferences. The demand for and preference toward cultural content often vary according to occupational background. For example, full-time workers may prioritize entertainment and relaxation due to work pressures and time constraints, while their interest in cultural identity-related content might be weaker. Thus, occupation plays an indispensable role in influencing media consumption behaviors and serves as a critical variable in MGA.

The moderating effect of occupation on consumer behavior regarding foreign films and TV series (see Table 7) indicates that occupational status significantly moderates several key variables and pathways ($CB_P$, $CB_{AME}$, $CV_0 \rightarrow CB_0$, $SPV_0 \rightarrow CB_0$, $SPV_0 \rightarrow DCI_{0,UPN} \rightarrow CB_0$, $CV_0 \rightarrow DCI_{0,IK} \rightarrow CB_0$, $CV_0 \rightarrow CTI_0 \rightarrow CB_0$, $CV_0 \rightarrow DCI_{0,UPN} \rightarrow CB_0$, and $SPV_0 \rightarrow CTI_0 \rightarrow CB_0$), such as the proportion of engagement, average monthly expenditure, and the relationships between perceived cultural values, practical and social values, and consumer behavior. The analysis provides partial support for Hypothesis 3d (**$H_{3d}$**).

Occupational status influences not only direct consumer behavior but also mediated relationships involving domestic cultural identity and cultural trade identity. Full-time workers, despite having limited leisure time, generally possess higher incomes, allowing them to be

**Table 7. Multi-group analysis of respondent's occupation.**

| | $C_{C,O,F}$ | $C_{C,O,O}$ | Difference |
|---|---|---|---|
| | **Mean** | | |
| $CB_P$ | 27.98% | 19.45% | 8.53%*** |
| $CB_{DE}$ | 2.32 hour | 2.36 hour | 0.04 |
| $CB_{AME}$ | RMB 65.89 | RMB 48.55 | 17.34* |
| | Path Coefficient | | |
| $CV_0 \rightarrow CB_0$ | 0.41*** | 0.83*** | -0.42*** |
| $SPV_0 \rightarrow CB_0$ | 0.27*** | -0.22** | 0.49*** |
| | Specific indirect effects | | |
| $SPV_0 \rightarrow DCI_{0,UPN} \rightarrow CB_0$ | 0.02 | 0.26*** | -0.24*** |
| $CV_0 \rightarrow DCI_{0,IK} \rightarrow CB_0$ | -0.03 | 0.05* | -0.09* |
| $CV_0 \rightarrow CTI_0 \rightarrow CB_0$ | 0.11*** | -0.03 | 0.14*** |
| $SPV_0 \rightarrow DCI_{0,IK} \rightarrow CB_0$ | -0.02 | 0.01 | -0.03 |
| $CV_0 \rightarrow DCI_{0,UPN} \rightarrow CB_0$ | 0.05 | -0.04 | 0.09* |
| $SPV_0 \rightarrow CTI_0 \rightarrow CB_0$ | 0.11*** | -0.03 | 0.14*** |

Note:
*** p < 1%,
** p < 5%,
* p < 10%.

more selective in their media consumption. These individuals may prioritize practical concerns due to their stressful and time-constrained schedules, which could diminish the impact of cultural values on their choices. Nevertheless, their financial capacity enables them to spend more on foreign media as a form of stress relief. Moreover, full-time workers often have a strong identification with their cultural trade identities and are sensitive to cultural shifts.

Conversely, part-time or other types of workers may engage less with foreign films and TV series, potentially due to a lesser interest in foreign cultures. Although they may have more leisure time, their engagement with media is less influenced by cultural trade identities. This discrepancy highlights the complex relationship between occupation and media consumption, underscoring the need to understand how cultural trade and domestic identity interact to shape consumer behavior.

Income level directly affects lifestyle, purchasing power, and individual priorities. High-income consumers typically possess greater social standing and broader access to cultural products, placing higher value on content quality and prestige, and often using foreign cultural products as a means of signaling social status. In contrast, lower-income consumers are more likely to focus on the utility and cost-effectiveness of cultural products. Therefore, income serves as a key variable in revealing differences in cultural consumption, particularly in terms of willingness to engage with and pay for foreign audiovisual content.

As detailed in Table 8, income significantly moderate several pathways in the model ($CB_{DE}$, $CB_{AME}$, $CV_0 \rightarrow CB_0$, $CV_0 \rightarrow DCI_{0,IK} \rightarrow CB_0$, $CV_0 \rightarrow CTI_0 \rightarrow CB_0$, $CV_0 \rightarrow DCI_{0,UPN} \rightarrow CB_0$, and $SPV_0 \rightarrow CTI_0 \rightarrow CB_0$), including their effects on average duration of engagement, average monthly expenditure, and the relationships between perceived cultural values, domestic cultural identity, and cultural trade identity.

Partial support for Hypothesis 3e indicates that while income do influence consumer engagement with foreign media, this impact is not uniform across all variables and pathways. Higher or lower income can affect how cultural values, trade identities, and domestic cultural

**Table 8. Multi-group analysis of respondent's income.**

| | $C_{C,I,H}$ | $C_{C,I,L}$ | Difference |
|---|---|---|---|
| | **Mean** | | |
| $CB_P$ | 20.76% | 23.76% | -3.00% |
| $CB_{DE}$ | 1.94 hour | 2.55 hour | -0.61*** |
| $CB_{AME}$ | RMB 66.60 | RMB 49.22 | 17.38* |
| | Path Coefficient | | |
| $CV_0 \rightarrow CB_0$ | 0.32*** | 0.72*** | -0.40*** |
| $SPV_0 \rightarrow CB_0$ | 0.22** | 0.04 | 0.18 |
| | Specific indirect effects | | |
| $SPV_0 \rightarrow DCI_{0,UPN} \rightarrow CB_0$ | 0.05 | 0.11*** | -0.06 |
| $CV_0 \rightarrow DCI_{0,IK} \rightarrow CB_0$ | 0.01 | 0.08*** | -0.07* |
| $CV_0 \rightarrow CTI_0 \rightarrow CB_0$ | 0.09 | -0.04 | 0.11* |
| $SPV_0 \rightarrow DCI_{0,IK} \rightarrow CB_0$ | 0.02 | 0.01 | 0.00 |
| $CV_0 \rightarrow DCI_{0,UPN} \rightarrow CB_0$ | 0.15** | 0.00 | 0.16** |
| $SPV_0 \rightarrow CTI_0 \rightarrow CB_0$ | 0.05 | -0.05* | 0.10** |

Note:
*** p < 1%,
** p < 5%,
* p < 10%.

identities interact to shape consumer preferences and engagement with foreign media. This complexity underscores the need for further research to elucidate these relationships and understand the specific ways in which income affect various aspects of consumer behavior in the context of foreign media consumption.

Consumers with low incomes often prioritize affordability, leading them to choose domestic over foreign media. In contrast, high-income consumers, who have access to a broader range of leisure activities, may view foreign media as less essential. High-income individuals might engage in 'conspicuous consumption,' using foreign media as a status symbol to signal their elevated social standing. Conversely, low-income consumers focus on fulfilling basic needs and do not use foreign media for status signaling. High-income consumers, with more leisure options, prioritize quality and prestige in their media choices, while low-income consumers emphasize practicality and cost-effectiveness. These differences highlight the intricate relationship between income, cultural trade, and domestic identity in shaping media consumption behaviors. Additional empirical studies are needed to further investigate these dynamics.

The analysis of income's impact on consumer behavior towards foreign films and TV series (as detailed in Table 8) demonstrates that income significantly moderate several key pathways ($CB_{DE}$, $CB_{AME}$, $CV_0 \rightarrow CB_0$, $CV_0 \rightarrow DCI_{0,IK} \rightarrow CB_0$, $CV_0 \rightarrow CTI_0 \rightarrow CB_0$, $CV_0 \rightarrow DCI_{0,UPN} \rightarrow CB_0$, and $SPV_0 \rightarrow CTI_0 \rightarrow CB_0$), including those affecting average duration of engagement, average monthly expenditure, and the relationships between perceived cultural values, domestic cultural identity, and cultural trade identity.

Partial support for Hypothesis 3e (**H$_{3e}$**) indicates that while income plays a role in shaping consumer engagement with foreign media, its effects are not uniform across all variables and pathways. This suggests that higher and lower income can influence how cultural values, trade identities, and domestic cultural identities interact to determine consumer choices and engagement with foreign media. The moderating effect of income highlights the complexity of

consumer behavior, revealing that income can impact the way individuals relate to and engage with media content.

High-income consumers often demonstrate different consumption patterns compared to low-income consumers. Low-income individuals generally prioritize affordability and are more likely to choose domestic over foreign media. High-income consumers, on the other hand, have access to a wider range of leisure activities and may perceive foreign media as less essential. They may engage in 'conspicuous consumption' of premium foreign media as a means of signaling their social status. Conversely, low-income consumers focus on fulfilling basic needs and do not use media as a status symbol. High-income individuals, with ample leisure options, emphasize quality and prestige in their media choices, while low-income individuals prioritize practicality and cost-effectiveness.

## Discussion

The relationship between cultural values and consumer behavior is significant because of the emotional and personal identity that is closely tied to cultural media content. This aligns with Hofstede's [17] Cultural Dimensions theory, which posits that cultural resonance has a deep emotional influence on consumer behavior. This finding also carries significant economic implications for the global media industry, underscoring the necessity for producers and marketers to integrate cultural diversity and sensitivity into foreign films and TV series. It is not merely the artistic quality that drives consumer engagement but the presentation of a wide range of cultural perspectives that fosters deeper audience connections [62]. On the other hand, the empirical data derived from this study does not conclusively establish a strong correlation between practical and social values and consumer choices, likely because these values are more utility-based and may not evoke strong emotional responses that drive consumer engagement. This outcome contrasts with existing theoretical frameworks on consumer behavior, such as consumer ethnocentrism and cosmopolitanism. These frameworks traditionally posit that consumers' choices are deeply influenced by their social identity, nationalistic tendencies, and openness to global products. However, the study's findings suggest a more nuanced relationship, where the influence of practical and social values may be less significant or subject to other mediating factors [63]. The results also suggest that the emotional and identity-based connections consumers form through cultural resonance are stronger drivers of behavior than the practical benefits of media consumption.

While consumer ethnocentrism typically leads to a preference for domestic goods, rooted in a sense of national pride, and cosmopolitanism encourages openness to international products, this research indicates that these constructs might not hold uniformly across different markets or product categories. Thus, the absence of strong empirical support challenges the prevailing assumptions, highlighting the need for a more context-specific analysis of consumer decision-making processes.

The findings reveal the significant mediating role of domestic cultural identity in shaping consumer preferences for foreign films and TV series [64]. The role is significant as it reflects how consumers prioritize their national and cultural pride when consuming foreign media, supporting findings from Zhao and Belk [65]. Cultural trade identity, in contrast, did not show a significant mediating effect, suggesting that while consumers may engage with global cultural trade, their consumption patterns are more strongly shaped by their domestic cultural ties rather than broader global trends.

Additionally, the study highlights the moderating effects of consumer demographics such as age, gender, and income on behavior, confirming that these characteristics significantly shape the causal and mediating relationships between perceived values of foreign films and TV

series and consumer engagement [66]. These insights offer valuable implications for marketers in this sector, emphasizing the importance of tailoring marketing strategies to align with consumer characteristics. The research offers a robust understanding of the dynamic interplay between consumer demographics and media consumption, urging foreign film and television enterprises to be adaptable to these variations for sustained economic success.

As indicated in Table 9, the study provides partial support for the hypothesis that cultural values, social values, and practical values of foreign films and TV series significantly influence consumer behavior. The emotional and identity-driven nature of cultural values emerged as a key determinant in shaping consumer preferences. This phenomenon is exemplified by the worldwide success of films like Crouching Tiger, Hidden Dragon, where cultural appeal transcended geographic boundaries, reflecting the deep resonance of cultural narratives [17].

Domestic cultural identity also played a critical mediating role in the relationship between these values and consumer behavior, highlighting the importance of cultural pride and emotional ties to one's native culture. In contrast, cultural trade identity was not found to have a significant mediating role, suggesting that the consumers' alignment with their domestic culture has a stronger influence than their engagement with the global cultural economy. This finding aligns with consumer behavior theory, which stresses the importance of native cultural connections in influencing media consumption. A pertinent case study is the Korean Wave (Hallyu), where South Korean films and television series gained international recognition while simultaneously reinforcing a strong sense of national cultural pride [65].

The study further identified that demographic factors such as age, education, and income moderate the relationship between these values and consumer behavior, particularly in shaping the depth and type of consumer engagement with foreign films and TV series. The finding supports the tenets of human capital theory. Individuals with higher education and greater income showed a stronger propensity to engage with foreign films and TV series, akin to their consumption of luxury foreign goods, often viewed as status symbols and reflections of cultural sophistication [43].

The study, situated within the Chinese context, provides critical economic insights into the influence of cultural values on consumers of foreign films and TV series [17]. It highlights the necessity for businesses in this industry to tailor marketing strategies that resonate with local cultural values. The findings demonstrate that cultural and domestic identities act as mediators between the cultural values of foreign films and TV series and consumer behavior, reinforcing the theories of Zhao and Belk [65]. This underscores the importance of culturally attuned marketing initiatives.

Additionally, the study shows that individual traits such as age, gender, and income moderate the relationship between cultural values and consumer behavior, supporting the work of Matzler et al. [67]. These insights suggest that personalized marketing approaches can be more effective in engaging diverse consumer segments. Furthermore, consumer engagement with foreign films and TV series is driven by an interplay of perceived cultural, social, and practical values, which marketers must consider to better connect with a broad range of consumers.

The role of demographic factors is also significant, as age, education, and income shape consumer behavior. For example, when respondents were asked about key Chinese cultural elements, traditional festivals (51.1%) and culinary practices (50.9%) were prioritized, followed by arts and crafts (49.4%) and regional music and dance (37.4%). These elements reflect the rich cultural heritage of China and present foreign films and TV series marketers with opportunities for deeper consumer engagement.

Moreover, recent data shows that 82.3% of respondents prefer online streaming platforms like Netflix and Hulu for consuming foreign films and TV series, highlighting the growing influence of digital media [23]. Though traditional outlets like television (56.4%) and cinemas

**Table 9. The hypotheses, findings, reasons, and related literature.**

| Hypothesis | Findings | Reasons | Literature |
|---|---|---|---|
| $H_1$: $CV_0$ and $SPV_0$ impact $CB_0$. | Supported partially: $CV_0$ significantly impacts $CB_0$, $SPV_0$ does not significantly impact $CB_0$. | Cultural values are closely tied to personal identity, driving emotional and subjective connections that significantly influence consumer behavior. According to Behavioral Economics Theory, consumers make decisions based not only on practical utility but also on cultural resonance and emotional satisfaction. This explains why perceived cultural values have a more substantial impact on consumer behavior compared to practical and social values. For instance, the success of international films like Crouching Tiger, Hidden Dragon in non-Asian markets illustrates how cultural appeal can transcend borders and deeply resonate with global audiences. The lack of significant influence from social and practical values can be explained by the argument that emotional engagement is a stronger driver than practical utility in media consumption. | Hofstede [17], Prince et al. [63] |
| $H_2$: $DCI_{0,UPN}$, $DCI_{0,IK}$ and $CTI_0$ mediate the relationship between $CV_0$, $SPV_0$, and $CB_0$. | Supported partially: $DCI_{0,UPN}$ and $DCI_{0,IK}$ significantly mediates, $CTI_0$ does not significantly mediate. | Consumers' domestic cultural identity plays a stronger role in mediating the relationship between cultural values and consumer behavior compared to cultural trade identity. This aligns with Consumer Behavior Theory, which posits that individuals with strong ties to their native culture are more likely to engage with content that aligns with their domestic identity. The Case of Korean Wave (Hallyu) shows how South Korean films and dramas have gained popularity globally while still reinforcing strong cultural pride domestically, further demonstrating the importance of cultural identity in shaping media consumption. | Zhao & Belk [65] |
| $H_3$: Consumer characteristics (age, gender, education, occupation, income) moderate the relationships between $CV_0$, $SPV_0$, $DCI_{0,UPN}$, $DCI_{0,IK}$, $CTI_0$, and $CB_0$. | Supported: Age, education, and occupation significantly moderate relationships. | Different demographic segments exhibit varied preferences based on cultural and practical values. Human Capital Theory explains that individuals with higher of education and income are better equipped to engage critically with foreign media content and derive cultural and intellectual value from it. This is evident in industries like luxury foreign goods consumption, where higher-income and educated consumers tend to show more openness to international products, perceiving them as status symbols that enhance their cultural capital. Similarly, foreign films and TV series are consumed differently by various demographic groups based on these factors. | Nicolaou & Kalliris [43] |

(29.1%) still hold relevance, the shift toward digital platforms is apparent. These platforms not only provide vast content libraries but also use algorithms for personalized recommendations, thus increasing user engagement [68]. The decreasing reliance on physical media, such as DVDs (8.3%), further emphasizes the convenience and accessibility offered by online streaming.

## Conclusions

The research investigates the influence of foreign films and TV series on the behavior of Chinese consumers, focusing on how demographic factors, such as age, gender, education, occupation, and income, influence consumer engagement with these media. The study is set against the backdrop of globalization and the proliferation of streaming platforms, which have facilitated increased cultural exchange and access to foreign media in China. The research also

explores the role of cultural trade and domestic cultural identity in shaping consumer preferences.

The primary objective of this research is to clarify the causal relationships between consumers' perceptions of the cultural, utilitarian, and societal values inherent in foreign films and TV series and their subsequent consumption patterns. The study aims to uncover the mediating roles played by cultural trade and domestic cultural identities, and to assess the moderating effects of individual characteristics on these relationships.

The research employs a mixed-methods approach, starting with a comprehensive literature review followed by a questionnaire survey to collect primary data from 786 respondents. Descriptive statistical analyses are used to analyze initial data, which are further refined using Partial Least Squares Structural Equation Modeling (PLS-SEM) to establish complex interrelationships among variables. Multi-group analysis (MGA) is also conducted to examine the moderating effects of demographic factors, including age, gender, education, occupation, and income, on consumer behavior.

The findings reveal that cultural values significantly influence consumer engagement with foreign films and TV series, whereas practical and social values do not exhibit the same level of impact. Domestic cultural identity serves as a significant mediator between cultural values and consumer behavior, whereas cultural trade identity does not have a notable mediating effect. Additionally, consumer demographics, such as age, gender, education, occupation, and income, significantly moderate the relationships between cultural values, cultural identity, and consumer engagement.

This study builds on existing consumer behavior theories by emphasizing the role of cultural values, domestic cultural identity, and social identity in shaping media consumption. It expands on Hofstede's [17] Cultural Dimensions theory and Zhao and Belk's [65] consumer behavior insights, showing that domestic cultural identity has a more substantial influence than cultural trade identity. The finding also challenges the assumptions that global cultural trade would significantly influence consumer patterns, instead highlighting the enduring power of national and cultural pride in consumers' engagement with global media content. This contributes to a broader framework of consumer behavior by highlighting the significance of emotional factors and personal identity (rather than merely social or practical considerations) in driving media consumption, particularly within a globalized media landscape.

For marketers and content creators, the study underscores the need for cultural sensitivity when producing and promoting foreign films and TV series. It shows that aligning content with domestic cultural identity, particularly in environments where cultural and personal values are closely linked, is crucial for engagement. Furthermore, demographic factors such as age, education, and income play critical roles in moderating consumer behavior, highlighting the necessity for media companies to develop more targeted and personalized campaigns. Overall, these findings demonstrate the importance of incorporating both cultural and demographic specificity into marketing strategies of foreign films and TV series.

The study contributes to the theoretical understanding of consumer behavior by integrating cultural hybridization theory, uses and gratifications theory, and theories of planned behavior to explain the complex dynamics of consumer engagement with foreign films and TV series. It advances the literature by highlighting the critical mediating role of domestic cultural identity and demonstrating the differential effects of demographic moderators on consumer behavior.

From a practical perspective, the study provides valuable insights for marketers and content creators aiming to penetrate the Chinese media market. It emphasizes the importance of cultural alignment and sensitivity in developing content that resonates with Chinese audiences. The findings suggest that producers and marketers should prioritize culturally resonant

themes and understand the unique preferences of different demographic groups to effectively engage the target audience and enhance market success.

The study's reliance on convenience sampling presents a key limitation, as it does not fully capture the population's diversity. This method can lead to demographic biases and limit the generalizability of the results. Additionally, the use of self-reported data introduces the possibility of response bias, where respondents may offer socially desirable answers rather than accurate reflections of their preferences. The study's cross-sectional design, which examines behavior at a single point in time, also limits the ability to assess behavioral changes over time or to establish causality between variables. Lastly, the study's focus on Chinese consumers restricts the global applicability of the findings, as cultural dynamics may differ in other regions.

To overcome these limitations, future studies should adopt a more diverse sampling method, such as stratified random sampling, to ensure broader representation. Longitudinal studies are recommended to track changes in consumer behavior over time, allowing for a more comprehensive understanding of how cultural values, cultural trade identity, and domestic cultural identity influence media consumption. Expanding the research to include respondents from various cultural backgrounds would also enhance the global applicability of the findings. Moreover, qualitative methods such as interviews and focus groups can provide deeper insights into the motivations and subjective experiences driving consumer engagement with foreign films and TV series. Lastly, the increasing role of AI and digital technologies in shaping consumer behavior, particularly in the context of AI-generated content in foreign films and TV series, offers a promising avenue for future research. Understanding the long-term impacts of these technological advancements on cultural consumption is essential for navigating the evolving media landscape.

## Supporting information

**S1 Table. Survey data.**
(XLSX)

**S1 Appendix. Questionnaire questions and results.**
(DOCX)

## Author Contributions

**Conceptualization:** Sheng Yuan, Hengbin Yin.

**Data curation:** Zhihan Chen.

**Formal analysis:** Wei-Lun Huang.

**Funding acquisition:** Sheng Yuan.

**Investigation:** Zhihan Chen.

**Methodology:** Wei-Lun Huang.

**Resources:** Zhihan Chen.

**Writing – original draft:** Wei-Lun Huang.

**Writing – review & editing:** Sheng Yuan.

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
