## [Decision Letter · Decision Letter 0]

22 Aug 2024

PONE-D-24-20331Unveiling the Mediating Role of Cultural Trade and Domestic Identity in Chinese Consumer Engagement with Foreign Films and TV SeriesPLOS ONE

Dear Dr. Yuan,

Thank you for submitting your manuscript to PLOS ONE. After careful consideration, we feel that it has merit but does not fully meet PLOS ONE’s publication criteria as it currently stands. Therefore, we invite you to submit a revised version of the manuscript that addresses the points raised during the review process.

**ACADEMIC EDITOR: **

The paper presents a detailed analysis of Chinese consumer behavior towards foreign movies and TV series (FFT) using a robust methodological framework. The results presented are interesting and relevant and the paper can be considered for publication after some corrections and considerations:

1- Revise the abstract, considering suppressing statistical results, and replacing "participants" with "respondents";

2. Revise the Introduction, considering including the full names of Foreign Movies and TV Series instead of FFT. Make clearer the importance of investigating customer engagement in Foreign Movies and TV Series. It is suggested that the introduction be organized to include the importance of this study, followed by the discussion of the constructs included in this study. This may include the discussion of mediating or moderating variables;

3. Revise the Literature Review, including the relevant theories, clearly indicating which estimation variables are linked to, discussing what are the theoretical gaps/contributions of this study. Try to rewrite this section in a more concise way, making the main arguments more accessible to a wider audience. It is also recommended to include a discussion of the relationship between CV and CTI, DCIupn and DCIik; SPV and CTI, DCIupn and DCIik; and CTI, DCIupn and DCIik and CB;

4. Revise the conceptual framework and methodology, separating the discussion of the measurement of variables; model used for estimation; research design and data collection procedures into different subsections. Make clearer the criteria for selecting interviewees (sampling technique) and how the minimum sample size was determined in this study. Include a discussion of the data collection procedure and measurement items. If higher-order PLS-SEM estimation is used, include the estimation procedure, as well as the multigroup analysis procedure. Include a detailed diagram of the conceptual framework;

5. Revise the discussion and conclusion section, including more clearly all the findings in the results; discuss why each of the relationships is significant/not significant; supported by what previous research, as well as the theoretical and practical implications of this study, as well as the limitations and recommendations for future research.

We look forward to receiving your revised manuscript.

Kind regards,

Angelo Marcelo Tusset

Academic Editor

PLOS ONE

Journal Requirements:

**Additional Editor Comments:**

The paper presents a detailed analysis of Chinese consumer behavior towards foreign movies and TV series (FFT) using a robust methodological framework. The results presented are interesting and relevant and the paper can be considered for publication after some corrections and considerations:

1- Revise the abstract, considering suppressing statistical results, and replacing "participants" with "respondents";

2. Revise the Introduction, considering including the full names of Foreign Movies and TV Series instead of FFT. Make clearer the importance of investigating customer engagement in Foreign Movies and TV Series. It is suggested that the introduction be organized to include the importance of this study, followed by the discussion of the constructs included in this study. This may include the discussion of mediating or moderating variables;

3. Revise the Literature Review, including the relevant theories, clearly indicating which estimation variables are linked to, discussing what are the theoretical gaps/contributions of this study. Try to rewrite this section in a more concise way, making the main arguments more accessible to a wider audience. It is also recommended to include a discussion of the relationship between CV and CTI, DCIupn and DCIik; SPV and CTI, DCIupn and DCIik; and CTI, DCIupn and DCIik and CB;

4. Revise the conceptual framework and methodology, separating the discussion of the measurement of variables; model used for estimation; research design and data collection procedures into different subsections. Make clearer the criteria for selecting interviewees (sampling technique) and how the minimum sample size was determined in this study. Include a discussion of the data collection procedure and measurement items. If higher-order PLS-SEM estimation is used, include the estimation procedure, as well as the multigroup analysis procedure. Include a detailed diagram of the conceptual framework;

5. Revise the discussion and conclusion section, including more clearly all the findings in the results; discuss why each of the relationships is significant/not significant; supported by what previous research, as well as the theoretical and practical implications of this study, as well as the limitations and recommendations for future research.

Reviewers' comments:

Reviewer's Responses to Questions

**Comments to the Author**

1. Is the manuscript technically sound, and do the data support the conclusions?

Reviewer #1: Yes

Reviewer #2: Yes

2. Has the statistical analysis been performed appropriately and rigorously? 

Reviewer #1: Yes

Reviewer #2: Yes

3. Have the authors made all data underlying the findings in their manuscript fully available?

Reviewer #1: Yes

Reviewer #2: Yes

4. Is the manuscript presented in an intelligible fashion and written in standard English?

Reviewer #1: Yes

Reviewer #2: Yes

5. Review Comments to the Author

Reviewer #1: It is an interesting research. However, some recommendations are included for the improvement of the manuscript.

1. Abstract

If it is survey type of study, the ‘participants’ should be ‘respondents’.

It is suggested that not to include statistic in the abstract. I.e. evidenced by an average monthly expenditure of RMB 141.37.

2. Introduction

It is suggested that to use the full name of Foreign Films and TV Series instead of FFT, unless FFT is a commonly accepted initial.

It would be good to include why it is critical to investigate customer engagement in Foreign Films and TV Series. i.e. why there is a need to conduct this study?

It is suggested that the introduction is organised to include the importance of this study (which includes the research gap or motivation of this study) followed by the discussion of the constructs included in this study. This could include the discussion of mediating or moderation variables.

3. Literature Review

It would be good to include the relevant theories in the literature review section. The theories should clearly indicate which variables of estimation are tied to. Also, discuss what are the theoretical gap/ contribution of this study.

It would be good to include the discussion of the relationship between CV to CTI, DCIupn and DCIik; SPV to CTI, DCIupn and DCIik; and CTI, DCIupn and DCIik to CB. If possible, use the subsection title to differentiate various relationships. For instance to arrange the relationship according to the subsections for “causal relationships between consumers' perceived values—cultural, practical, and social—of FFT and their subsequent behaviors. It also investigates the mediating roles of CTI0 and DCI0, as well as the moderating effects of consumer characteristics.”

4. Conceptual Framework and Methodology

It would be good to separate the discussion of measurement of the variables; model used for estimation; research design and data collection procedures into different subsections.

If it is a survey, the “respondents” should be referred to instead of “participants”. What are the respondents’ selection criteria (sampling technique) and how the minimum sample size was determined in this study?

The discussion of the procedure of data collection and measurement items are missing.

Does higher order PLS-SEM estimation is used? If so, please include the procedure of estimation in this section. Also, include the multi-group analysis procedure.

5. Discussion and Conclusion

The discussion should include all the findings of in the results; discuss why each of the relationship is significant/ not significant; supported by which previous research.

Also include the theoretical and practical implications of this study as well as the limitations and recommendations of future research.

Reviewer #2: Your paper provides an in-depth analysis of Chinese consumer behavior towards foreign films and TV series (FFT) using a robust methodological framework. The study's exploration of cultural trade and domestic identity as mediating factors is a significant contribution to the literature on cultural consumption and consumer behavior in China. I’m including below some strengths and weaknesses (and potential areas for improvement).

Strengths:

- The manuscript includes an extensive review of existing literature, which sets a strong foundation for the research, it effectively situates the study within the broader context of cultural trade and consumer behavior.

- The use of PLS is appropriate for the complex relationships being examined; the study's methodological approach is well-executed and aligns with the research objectives.

- The focus on the mediating roles of cultural trade and domestic cultural identity in FFT consumption is innovative and addresses a gap in the current literature.

- The collection of primary data from 786 participants and the detailed analysis using descriptive statistics and PLS-SEM provide a robust empirical foundation for the study's findings.

- The manuscript offers theoretical contributions by integrating cultural trade and domestic identity into the understanding of consumer behavior, which can have practical implications for marketers and content creators.

Some suggestions for improvement:

- Some sections of the manuscript, particularly the literature review and conceptual framework, are dense and could benefit from more concise writing; streamlining these sections would improve readability and make the key arguments more accessible to a broader audience.

- The manuscript should avoid overly technical language where possible. So, try to simplify some of the statistical explanations to make the research more approachable to readers who may not be familiar with advanced SEM techniques.

- While the conceptual framework is well-developed, it could be presented more clearly in the manuscript, so the inclusion of a detailed diagram or visual representation of the framework could help readers better understand the relationships between the variables.

- The discussion section should more explicitly connect the findings to the research questions and hypotheses, to clarify how the results contribute to answering the study's primary questions.

- There is room for a more critical evaluation of the findings in relation to existing literature. The manuscript could benefit from a deeper discussion on how the results compare to similar studies and what this means for the field.

6. PLOS authors have the option to publish the peer review history of their article (what does this mean?). If published, this will include your full peer review and any attached files.

Reviewer #1: No

Reviewer #2: No

---

## [Author Response · Author response to Decision Letter 0]

1 Oct 2024

Dear editor and reviewers,

Thank you very much for your valuable comments and suggestions, based on which we have performed a very thorough revision on the manuscript. As a result, tracked changes (highlighted in red) are almost everywhere throughout the manuscript; however, the basic structure and key ideas remain intact. We hope you will find the revision satisfactory. The following are our responses to your comments. Thank you!

Sincerely,

Authors

Responses to Academic Editor’s Comments

The paper presents a detailed analysis of Chinese consumer behavior towards foreign movies and TV series (FFT) using a robust methodological framework. The results presented are interesting and relevant and the paper can be considered for publication after some corrections and considerations:

Point 1: Revise the abstract, considering suppressing statistical results, and replacing "participants" with "respondents".

Response 1: Thank you for your valuable suggestion to revise the abstract. We have removed specific statistical results (such as the average monthly expenditure of RMB 141.37) to make the abstract (p. 1) more generalized and academic. The terms of “participants” have been replaced with “respondents” throughout the manuscript. 

Point 2: Revise the Introduction, considering including the full names of Foreign Movies and TV Series instead of FFT. Make clearer the importance of investigating customer engagement in Foreign Movies and TV Series. It is suggested that the introduction be organized to include the importance of this study, followed by the discussion of the constructs included in this study. This may include the discussion of mediating or moderating variables.

Response 2: Thank you for your valuable suggestion to revise the Introduction. Here is the one-to-one correspondence of questions and answers based on the introduction revision you requested:

1. Replacing "FFT" with "Foreign Movies and TV Series"

Answer: Yes, all instances of "FFT" have been replaced with "foreign films and TV series," and the full term is used consistently throughout the manuscript.

2. Emphasizing the importance of investigating consumer engagement with Foreign Movies and TV Series and highlighting the importance of the study

o Answer: Yes, the revised introduction (p. 2-5) opens with a discussion on the importance of foreign films and TV series in globalization and cultural exchange. Then, it clearly states the significance of understanding consumer engagement with foreign films and TV series for both industry practices and academic research. 

3. Discussing the constructs, mediating and moderating variables included in the study

o Answer: Yes, the revised introduction (p. 2-5) outlines the study's focus on examining consumers' perceptions of the cultural, utilitarian, and societal values of foreign films and TV series and how these perceptions influence their consumption patterns. It also discusses the mediating roles of cultural trade and domestic cultural identities, as well as the moderating effects of individual characteristics.

The Introduction has been revised as follows (the content highlighted in red is the response to your suggestions, and the italicized explanations in parentheses have been removed from the revised manuscript):

Globalization has profoundly intensified cultural exchanges, particularly through the proliferation of foreign films and TV series (replacing FFT) [1-3]. The United Nations has documented the rapid expansion of global trade in creative goods, including foreign films and TV series, which is growing at a pace that surpasses traditional merchandise trade [4, 5]. This trend is largely driven by the widespread adoption of digital technologies and the increasing availability of internet access. Against this backdrop, streaming giants like Netflix now possess a very extensive and diverse subscriber base across multiple continents. These platforms not only extend their business reach globally but also act as pivotal agents of cultural diplomacy, enhancing global cultural literacy and promoting cross-cultural dialogues, especially in the realm of films and TV series (discussing the background of the study: the importance of foreign films and TV series in globalization and cultural exchange) [1, 6-8]. Thus, detailed investigations into consumer engagement with foreign films and TV series can yield vital data on viewing preferences and behaviors, providing practical advice for media marketers and content producers engaging a global audience. Moreover, China’s status as the world’s second largest media market [9] underscores the critical need to conduct research on its audiences to generate China-specific insights for international marketers (emphasizing the importance of the study for industry practitioners).

In addition to industry practice, a comprehensive understanding of consumer engagement with foreign films and TV series is also essential for academic inquiry [10] (emphasizing the importance of the study for academia). The theoretical frameworks employed in consumer behavior research offer valuable insights into the effects of engagement with foreign films and TV series. Cultural hybridization theory suggests that these media products serve as a crucible for the blending of diverse cultural norms, leading to the creation of new cultural identities. Similarly, uses and gratifications theory posits that individuals engage with these media products to satisfy specific personal needs, such as entertainment, education, and cultural enrichment. Further, theories such as the theory of planned behavior and the theory of reasoned action elucidate how attitudes, social norms, and perceived control influence individual preferences for foreign films and TV series from different cultural backgrounds (discussion of constructs, including mediating and moderating variables). Collectively, these theories suggest that individual attitudes and interpretations of cultural values play a significant role in shaping consumption patterns [11, 12]. However, although a number of studies [13-16] have explored this role, very few have comprehensively examined the complexities of consumer engagement with foreign films and TV series, particularly considering a range of mediating and moderating variables (pointing out the research gap). 

Motivated by this research gap (stating the motivation of this study), the current study adopts a robust methodological framework, incorporating literature reviews, surveys, descriptive statistical analyses, and Partial Least Squares Structural Equation Modeling (PLS-SEM) to scrutinize consumer behavior concerning foreign films and TV series within the Chinese context. The primary objective of this research is to clarify the causal relationships between consumers' perceptions of the cultural, utilitarian, and societal values inherent in these media products and their subsequent consumption patterns (importance of this study and constructs). Central to this investigation is the analysis of the mediating roles played by cultural trade and domestic cultural identities (discussion of mediating variables) in shaping these behaviors, alongside an assessment of the moderating effects exerted by individual characteristics (discussion of moderating variables). This approach rigorously addresses the multifaceted and dynamic nature of consumer behavior.

The study makes significant theoretical contributions across several critical domains: First, it deepens our understanding of the determinants influencing the consumption of foreign films and TV series in China. Second, it fills a crucial gap in existing literature by exploring the mediating roles of cultural trade and domestic cultural identities. Third, it adds a layer of complexity to consumer behavior research by incorporating individual attributes as moderating factors. Fourth, it offers unique insights specific to China, illuminating the distinct cultural, economic, and social dynamics within the region. Collectively, these contributions present a sophisticated, multi-dimensional framework for comprehending the consumption of foreign films and TV series in China, thereby broadening the study's relevance to various cultural contexts and enhancing both theoretical and practical knowledge.

The article is systematically organized into clearly defined sections, each dedicated to a specific aspect of the investigation into the impact of foreign films and TV series consumption on consumer behavior and cultural identity. The second section delivers an exhaustive literature review, covering seminal works on these subjects [17-22]. The third section delineates the study's theoretical framework, identifying crucial variables such as the perceived cultural, practical, and social values of foreign films and TV series, as well as consumer behavior, in conjunction with mediating and moderating variables like cultural trade and individual characteristics.

The fourth section elaborates on the research methodology, detailing the study design, data collection techniques, and analytical methods employed. It also presents findings derived from descriptive statistics and PLS-SEM, establishing causal links between perceived values of foreign films and TV series and consumption behaviors while examining the mediating roles of cultural trade and domestic identities. The fifth section interprets the results of multigroup analysis, emphasizing the moderating effects of individual attributes. The sixth section critically assesses the findings in relation to the hypotheses, while also recognizing the study's limitations. The final section summarizes the key findings, discusses their theoretical and practical implications, and proposes avenues for future research.

Point 3: Revise the Literature Review, including the relevant theories, clearly indicating which estimation variables are linked to, discussing what are the theoretical gaps/contributions of this study. Try to rewrite this section in a more concise way, making the main arguments more accessible to a wider audience. It is also recommended to include a discussion of the relationship between CV and CTI, DCIupn and DCIik; SPV and CTI, DCIupn and DCIik; and CTI, DCIupn and DCIik and CB;

Response 3: Thank you for your valuable suggestion to revise the Literature Review. Here is a detailed one-to-one response to the specific questions you provided:

1. Question: Does the Literature Review include relevant theories?

Answer: Yes, the Literature Review incorporates theories such as Globalization Theory (p. 6, line 114-120), Cultural Hybridization (p. 8, line 123-128), and Technological Convergence (p. 7-8, line 129-136) to explain the influence of foreign movies and TV series on global consumer behavior.

2. Question: Are the estimation variables clearly linked?

Answer: Yes, the Literature Review clearly links cultural values (CV), cultural trade influences (CTI), domestic cultural identities (DCIupn and DCIik), and social prestige values (SPV) with consumer behavior (CB). (p. 8, line 145-147, p. 9, line 159-161)

3. Question: Does the Literature Review discuss the theoretical gaps and contributions of the study?

Answer: Yes, the Literature Review identifies gaps in understanding the mediating roles of cultural trade and domestic cultural identity, and the moderating effects of consumer characteristics (p. 8, line 147-149). The study contributes by filling these gaps, offering new insights into the complex dynamics of global media consumption (from p. 11, line 210 to p. 12 line 223).

4. Question: Is the Literature Review concise and accessible to a wider audience?

Answer: Yes, the section has been rewritten to be more concise, focusing on the key arguments and making the main points more accessible, ensuring clarity without sacrificing academic rigor (p. 5-12).

5. Question: Does the Literature Review include a discussion of the relationships between CV and CTI, DCIupn and DCIik; SPV and CTI, DCIupn and DCIik; and CTI, DCIupn and DCIik and CB?

Answer: Yes, the Literature Review discusses how these variables interact, particularly focusing on the mediating and moderating roles they play in shaping consumer behavior towards foreign movies and TV series. The relationships are thoroughly explored, providing a deeper understanding of their interconnections (from p. 8 line 150 to p. 11 line 209).

Point 4: Revise the conceptual framework and methodology, separating the discussion of the measurement of variables; model used for estimation; research design and data collection procedures into different subsections. Make clearer the criteria for selecting interviewees (sampling technique) and how the minimum sample size was determined in this study. Include a discussion of the data collection procedure and measurement items. If higher-order PLS-SEM estimation is used, include the estimation procedure, as well as the multigroup analysis procedure. Include a detailed diagram of the conceptual framework;

Response 4: Thank you for your valuable suggestion to revise the conceptual framework and methodology. Here is a detailed one-to-one response to the specific questions you provided:

Questions and Answers Based on Point 4:

1. Revise the conceptual framework and methodology, separating the discussion of the measurement of variables; model used for estimation; research design and data collection procedures into different subsections.

Answer: In the revised conceptual framework and methodology, we provided clear distinctions among the measurement of variables (p. 12-20), model estimation (p. 21-23), and research design and data collection procedures (p. 23-27). Specifically, the measurement of variables subsection (p. 12-20) focused on the metrics that were used to assess consumer behavior towards foreign films and TV series. These include the proportion of engagement (measured by average viewing time and financial expenditure) and perceptions of cultural and social values. The model used for estimation subsection (p. 21-23) centered around Partial Least Squares Structural Equation Modeling (PLS-SEM), which is particularly effective in handling complex relationships among latent variables. The research design and data collection procedures subsection (p. 23-27) elaborated on the use of a structured survey distributed to Chinese consumers, targeting their interaction with foreign media content.

2. Make clearer the criteria for selecting interviewees (sampling technique) and how the minimum sample size was determined in this study.

Answer: Respondents were selected using convenience sampling, which targeted habitual Chinese viewers of foreign films and TV series. This non-probability sampling method was employed due to the specific focus on participants who were readily available and willing to participate in the study (from p. 25 line 474 to p. 26 line 484). To determine the minimum sample size, we used general guidelines from PLS-SEM literature, which recommend a sample size of at least ten times the number of indicators in the most complex construct. In this case, a total of 786 valid responses were collected, which exceeded the minimum threshold required for robust analysis in this study (p. 26 line 485-494).

3. Include a discussion of the data collection procedure and measurement items.

Answer: Data collection was conducted via an online survey, targeting Chinese consumers of foreign films and TV series. The survey was carefully constructed to gather data on key variables, and was conducted after obtaining a waiver for ethical approval (p. 25, line 467-473). The measurement items were designed based on established scales from prior research and included Likert-type questions to assess the degree of engagement, perception of cultural trade, and domestic cultural identity (from p. 12 line 225 to p. 20 line 390). The collected data was then cleaned and processed for analysis using the PLS-SEM framework (p. 26, line 495-496).

4. If higher-order PLS-SEM estimation is used, include the estimation procedure, as well as the multigroup analysis procedure.

Answer: The estimation procedure involved the use of higher-order PLS-SEM, with SmartPLS software being utilized for model estimation. This included both the evaluation of the measurement model (assessing reliability and

---

## [Decision Letter · Decision Letter 1]

18 Oct 2024

PONE-D-24-20331R1Unveiling the Mediating Role of Cultural Trade and Domestic Identity in Chinese Consumer Engagement with Foreign Films and TV SeriesPLOS ONE

Dear Dr. Yuan,

Thank you for submitting your manuscript to PLOS ONE. After careful consideration, we feel that it has merit but does not fully meet PLOS ONE’s publication criteria as it currently stands. Therefore, we invite you to submit a revised version of the manuscript that addresses the points raised during the review process.

**ACADEMIC EDITOR: **

The authors conducted a comprehensive review of the paper, submitting a version that could be accepted for publication after minor considerations:

1- Include the nomenclature of each variable in Table 1;

2- Include a table with the questions from the questionnaire in an appendix to the paper;

3- Justify the choice of variables, age, gender and education, for the MGA analysis;

4- Include after Figure 4, whether the hypothesis was accepted or rejected;

5- Consider a discussion section and one just for the conclusion;

6- Make clearer in the conclusions what the contributions of the study are to theoretical and practical aspects, and what the limitations and recommendations for future studies are.

We look forward to receiving your revised manuscript.

Kind regards,

Angelo Marcelo Tusset

Academic Editor

PLOS ONE

Journal Requirements:

Additional Editor Comments:

The authors conducted a comprehensive review of the paper, submitting a version that could be accepted for publication after minor considerations:

1- Include the nomenclature of each variable in Table 1;

2- Include a table with the questions from the questionnaire in an appendix to the paper;

3- Justify the choice of variables, age, gender and education, for the MGA analysis;

4- Include after Figure 4, whether the hypothesis was accepted or rejected;

5- Consider a discussion section and one just for the conclusion;

6- Make clearer in the conclusions what the contributions of the study are to theoretical and practical aspects, and what the limitations and recommendations for future studies are.

Reviewers' comments:

Reviewer's Responses to Questions

**Comments to the Author**

1. If the authors have adequately addressed your comments raised in a previous round of review and you feel that this manuscript is now acceptable for publication, you may indicate that here to bypass the “Comments to the Author” section, enter your conflict of interest statement in the “Confidential to Editor” section, and submit your "Accept" recommendation.

Reviewer #1: (No Response)

Reviewer #2: All comments have been addressed

2. Is the manuscript technically sound, and do the data support the conclusions?

Reviewer #1: Yes

Reviewer #2: Yes

3. Has the statistical analysis been performed appropriately and rigorously? 

Reviewer #1: Yes

Reviewer #2: Yes

4. Have the authors made all data underlying the findings in their manuscript fully available?

Reviewer #1: Yes

Reviewer #2: Yes

5. Is the manuscript presented in an intelligible fashion and written in standard English?

Reviewer #1: Yes

Reviewer #2: Yes

6. Review Comments to the Author

Reviewer #1: 1. Table 1 has been included to explain each of the variables. It would be good to indicate the full name for each of the variables in Table 1. Also, surveys are used to collect data from the respondents. What are the questions for the questionnaire? It would be good to provide the questions (Instruments or items) and the source of the questions in a sperate table or combine with Table 1.

2. For the interpretation such as in page 34 (the writing after Figure 4) does not mention whether the Hypothesis is supported or not supported.

3. Any hypothesis proposed for the analysis of MGA? Why only choose age, gender and education for the MGA analysis?

4. Please separate the Discussion Section with Conclusion Section. For discussion, it would be good to discuss the overall analysis (such as in Figure 2), followed by MGA for age, gender and education. All of the discussion could be supported by the findings of previous studies.

5. What are the implications of this study from theoretical and practical aspects? What are the limitations and recommendations of future study?

Reviewer #2: Overall, the authors have improved the manuscript significantly, and it includes more information about its findings and contributions.

7. PLOS authors have the option to publish the peer review history of their article (what does this mean?). If published, this will include your full peer review and any attached files.

Reviewer #1: No

Reviewer #2: No

---

## [Author Response · Author response to Decision Letter 1]

30 Oct 2024

Response to ACADEMIC EDITOR Comments

The authors conducted a comprehensive review of the paper, submitting a version that could be accepted for publication after minor considerations:

Point 1: Include the nomenclature of each variable in Table 1

Response 1: 

Thank you for your valuable feedback on our manuscript. We appreciate your insights and suggestions for improving the clarity and rigor of our work.

In response to your comment, we have included the nomenclature of each variable in Table 1 to ensure that all constructs and variables are clearly defined and easy to understand for readers. This addition aims to enhance the comprehensibility of our study and provide a more structured overview of the variables used in our analysis.

“In this study, the variables were measured meticulously to capture the intricate aspects of consumer behavior towards foreign films and TV series, ensuring that each dimension was analyzed with precision and depth. (see Table 1 and Appendix A)

Table 1. Nomenclature of Dimensions and Constructs of Consumer Behavior and Cultural Perceptions in Foreign Films and TV Series

Dimension Variable Symbol Construct Description

Consumer Behavior ({\\mathbf{CB}}_\\mathbf{0})

 {\\rm CB}_P Engagement with foreign content, capturing frequency and enthusiasm for watching foreign films and TV series.

 {\\rm CB}_{DE} Depth of commitment, indicating the emotional and time-related investment in foreign content.

 {\\rm CB}_{AME} Economic and perceived value, reflecting financial expenditure and personal valuation of foreign entertainment.

Cultural Value ({\\mathbf{CV}}_\\mathbf{0}) {\\rm CV}_{CP} Diversity recognition, highlighting the awareness and appreciation of varied cultures presented in media.

 {\\rm CV}_{CC} Cultural learning tool, where foreign content is seen as a medium for understanding different cultural norms.

 {\\rm CV}_{GCI} Global cultural exchange facilitator, promoting intercultural communication through content.

 {\\rm CV}_{SHC} Sociopolitical context, using content to gain insight into international sociopolitical environments.

 {\\rm CV}_{TUR} Understanding and respect, fostering empathy and respect for different cultures through foreign media.

Social and Practical Values ({\\mathbf{SPV}}_\\mathbf{0}) {\\rm SPV}_{LSCE} Linguistic and cultural exchange value, assessing how content aids in cross-linguistic and cultural understanding.

 {\\rm SPV}_{ICS} Cross-cultural communication skills, recognizing the role of media in enhancing interaction across cultures.

 {\\rm SPV}_{AD} Artistic appreciation, reflecting the cultural and artistic enrichment from media consumption.

 {\\rm SPV}_{EW} Societal understanding, encouraging a broadened perspective of other societies.

 {\\rm SPV}_{PE} Personal growth and ethical development, assessing the influence on an individual's values and ethics.

Cultural Trade Identities ({\\mathbf{CTI}}_\\mathbf{0}) {\\rm CTI}_{GCI} International dialogue and diversity, emphasizing the promotion of cultural diversity through global trade.

 {\\rm CTI}_{CH} Cultural preservation, underlining the importance of maintaining and promoting national cultures through trade.

 {\\rm CTI}_{EA} Economic amplification, stressing cultural trade's role in economic growth.

 {\\rm CTI}_{ACI} Cultural industry development, focusing on expanding cultural industries through trade.

 {\\rm CTI}_{ECC} Export sustainability, considering the sustainability of cultural trade practices.

Domestic Cultural Identity ({\\mathbf{DCI}}_\\mathbf{0}) {\\rm DCI}_{CU} Unique cultural traits, highlighting distinctive national cultural features.

 {\\rm DCI}_{CP} Cultural pride, reflecting individuals' pride in their national culture.

 {\\rm DCI}_{UN} Cultural understanding, indicating deep personal knowledge of one’s own culture.

 {\\rm DCI}_{CK} Cultural knowledge transmission, signifying the sharing and inheritance of cultural knowledge.

 {\\rm DCI}_{CI} Cultural influence, focusing on how a nation’s culture impacts global awareness.

Consumer Characteristics ({\\mathbf{CC}}_\\mathbf{0}) {\\rm CC}_A Age as a factor, assessing how age influences content choices.

 {\\rm CC}_G Gender impact, examining how gender affects preferences and communication regarding media consumption.

 {\\rm CC}_E Education shaping values, exploring how educational background shapes personality, values, and preferences.

 {\\rm CC}_O Occupation influence, considering how occupation-related skills and behaviors affect media choices.

 {\\rm CC}_I Income effect, reflecting how income influences lifestyle and media consumption priorities.

Please let us know if there are any other aspects that require further clarification or if you have additional suggestions.

Point 2: Include a table with the questions from the questionnaire in an appendix to the paper

Response 2: 

Thank you very much for your constructive feedback on our manuscript. We sincerely appreciate your detailed review and recommendations for improving the quality of our work.

In response to your suggestion, we have now included a table with the questions from the questionnaire in an appendix to the paper. This addition aims to provide transparency and allow readers to fully understand the measures and constructs used in our study. We believe this enhancement will contribute significantly to the replicability and overall clarity of our research.

“Appendix A: Questionnaire Questions and Results

 Question

1 How often do you engage with foreign films and TV series? 

2 How would you rate your emotional connection to foreign content? 

3 How much do you spend monthly on foreign films and TV series? 

4 Do you believe foreign films help in recognizing cultural diversity?

5 How much does foreign content influence your cultural practices? 

6 Do you view foreign content as a means for intercultural exchange?

7 Does foreign content help you understand international sociocultural contexts? 

8 Do you feel foreign films promote respect for other cultures? 

9 How valuable do you find foreign films in terms of foreign language skill learning? 

10 Do foreign films improve your cross-cultural sensitivity? 

11 How much do you appreciate the artistic value in foreign content?

12 Do foreign films encourage you to have a broader view of different societies? 

13 How do foreign films influence your personal growth and ethical views? 

14 How important is intercultural economic exchange to you? 

15 How do foreign films contribute to cultural diversity promotion?

16 How does cultural trade impact economic growth? 

17 Do you consider foreign content as a factor in cultural industry development? 

18 How important is cultural preservation and expansion in the context of global media? 

19 How much pride do you have in your cultural heritage when engaging with foreign media? 

20 Does foreign content help you understand your national culture better?

21 How does engaging with foreign content influence your perception of your culture globally? 

22 What is your age?

23 What is your gender?

24 What is your highest education? 

25 What is your occupation?

26 What is your annual income?

”

Please feel free to let us know if there are any additional changes that you would recommend.

Point 3: Justify the choice of variables, age, gender and education, for the MGA analysis

Response 3: 

Thank you very much for your insightful feedback and suggestions to improve our manuscript. We greatly appreciate your careful review.

Regarding your comment on justifying the choice of variables—age, gender, and education—for the MGA analysis, we have provided additional justification in the revised manuscript. Specifically, these variables were selected due to their well-established influence on consumer behavior in the context of foreign media consumption. Age affects openness to foreign cultural influences, as younger individuals are generally more exposed to global media through digital platforms. Gender influences media preferences, with different types of content resonating variably across male and female audiences. Education impacts cultural cognition, as individuals with higher educational attainment often show greater critical thinking abilities and a deeper appreciation for foreign cultural content. These factors make age, gender, and education crucial moderators in understanding the differentiated consumer engagement with foreign films and TV series.

“Age is a key determinant of consumer behavior, as individuals across different age groups display divergent value systems, beliefs, and priorities. These differences substantially affect their acceptance of foreign cultural content. Younger consumers are generally more open to diverse cultural influences, largely due to their exposure to globalization and their access to foreign cultures through digital platforms. Conversely, older consumers tend to favor content that aligns closely with their established cultural backgrounds. Hence, age serves as a significant factor in determining both the acceptance of and engagement with foreign films and television. The MGA in this study effectively highlights the pronounced distinctions across different age groups.

…

Gender influences individual preferences and communication styles, thereby affecting engagement with foreign audiovisual content. For instance, men and women may exhibit distinct preferences regarding media genres, with men often favoring action or sports-related content, while women may prefer drama or romance. Additionally, cultural roles and societal expectations linked to gender may moderate the degree of acceptance of foreign content. Thus, gender functions as a moderating variable that aids in comprehensively understanding the differences in the consumption of foreign cultural products.

…

Educational attainment significantly impacts an individual's value system, cultural cognition, and media consumption behavior. Higher education are generally correlated with greater critical thinking abilities and cultural appreciation, enabling individuals to understand foreign cultural content more profoundly, including its social and economic dimensions. Highly educated consumers tend to show a greater acceptance and appreciation for foreign films and television due to their capacity to derive cultural and intellectual value from such content.

…

Occupation determines not only an individual's economic capacity but also shapes personality traits and lifestyle preferences. The demand for and preference toward cultural content often vary according to occupational background. For example, full-time workers may prioritize entertainment and relaxation due to work pressures and time constraints, while their interest in cultural identity-related content might be weaker. Thus, occupation plays an indispensable role in influencing media consumption behaviors and serves as a critical variable in MGA.

…

Income level directly affects lifestyle, purchasing power, and individual priorities. High-income consumers typically possess greater social standing and broader access to cultural products, placing higher value on content quality and prestige, and often using foreign cultural products as a means of signaling social status. In contrast, lower-income consumers are more likely to focus on the utility and cost-effectiveness of cultural products. Therefore, income serves as a key variable in revealing differences in cultural consumption, particularly in terms of willingness to engage with and pay for foreign audiovisual content.

”

We hope this justification clarifies our approach, and we are open to further comments or suggestions you may have.

Point 4: Include after Figure 4, whether the hypothesis was accepted or rejected

Response 4: 

Thank you for your constructive feedback and for pointing out areas for improvement. We truly value your input in enhancing the clarity and rigor of our manuscript.

In response to your suggestion, we have now included statements after Figure 4 indicating whether each hypothesis was accepted or rejected. Specifically, we have clearly stated the results of the hypothesis testing, based on the path coefficients and their statistical significance levels. This addition aims to improve the reader's understanding of the outcomes of our analysis and ensures that the conclusions drawn from the data are communicated transparently.

“The PLS-SEM analysis provides additional insights into Chinese consumer behavior in relation to foreign films and TV series (see Figure 4). Hypothesis 1 (H_1) receives partial support, with a strong positive relationship observed between perceived cultural values and consumer behavior towards foreign films and TV series ({\\rm CV}_0 → {\\rm CB}_0), with a path coefficient of 0.60. This suggests that cultural consumption allows consumers to express their identities and values, offering them a platform to explore and integrate diverse cultural paradigms. However, the relationship between perceived practical and social values and consumer behavior ({\\rm CV}_0 → {\\rm CB}_0) was found to be statistically insignificant. This could indicate that while cultural values are closely tied to personal identity and provide a meaningful emotional experience, practical and social values may not have the same influence on consumer behavior, as they do not offer the same depth of personal enrichment or emotional engagement.

 mean

{\\rm CV}_0 → {\\rm CTI}_0 → {\\rm CB}_0 0.01

{\\rm CV}_0 → {\\rm DCI}_{0,\\ UPN} → {\\rm CB}_0 0.04*

{\\rm CV}_0 → {\\rm DCI}_{0,\\ IK} → {\\rm CB}_0 0.03*

{\\rm SPV}_0 → {\\rm CTI}_0 → {\\rm CB}_0 0.02

{\\rm SPV}_0 → {\\rm DCI}_{0,\\ UPN} → {\\rm CB}_0 0.13***

{\\rm SPV}_0 → {\\rm DCI}_{0,IK} → {\\rm CB}_0 0.01

Fig 4. The results of PLS-SEM analysis.

Note: *** p < 1%, ** p < 5%, * p < 10%.

Hypothesis 2 (H_2) is substantiated to a degree, as evidenced by the statistically significant positive mediating effects of the emotional and subjective dimensions of domestic cultural identity on both cultural values and practical and social values in shaping consumer behavior towards foreign films and TV series. Specifically, the indirect effects in the pathways from cultural values to domestic cultural identity to consumer behavior, and from practical and social values to domestic cultural identity to consumer behavior ({\\rm CV}_0 → {\\rm DCI}_{0,\\ UPN} → {\\rm CB}_0 and {\\rm SPV}_0 → {\\rm DCI}_{0,\\ UPN} → {\\rm CB}_0), are measured at 0.04 and 0.13, respectively. Furthermore, a notable positive indirect effect is observed in the pathway from cultural values to consumer behavior through the knowledge-based dimension of domestic cultural identity ({\\rm CV}_0 → {\\rm DCI}_{0,IK} → {\\rm CB}_0), quantified at 0.03.”

We hope this addresses your comment effectively, and we are more than willing to make further revisions if necessary.

Point 5: Consider a discussion section and one just for the conclusion

Response 5: 

Thank you very much for your valuable feedback and thoughtful suggestions on our manuscript. We are grateful for your insights, which have greatly contributed to improving our work.

In response to your suggestion, we have now restructured the manuscript to include distinct sections for the discussion and the conclusion. The discussion section focuses on interpreting the findings in light of existing literature, exploring the implications, and providing a critical analysis of the results. The conclusion section succinctly summarizes the key findings, outlines the contributions of the study, and provides suggestions for future research. This separation aims to enhance the clarity and coherence of our argument, ensuring that readers can easily differentiate between the interpretation of findings and the final summarizing statements.

We hope that these modifications meet your expectations, and we are open to any further recommendations you may have.

Point 6: Make clearer in the conclusions what the contributions of the study are to theoretical and practical aspects, and what the limitations and recommendations for future studies are.

Response 6: 

Thank you for your insightful feedback and for helping us improve the clarity of our manuscript. We greatly value your suggestions.

In response to your comment, we have revised the conclusions section to clearly delineate the theoretic

---

## [Decision Letter · Decision Letter 2]

11 Nov 2024

Unveiling the Mediating Role of Cultural Trade and Domestic Identity in Chinese Consumer Engagement with Foreign Films and TV Series

PONE-D-24-20331R2

Dear Dr. Wei-Lun Huang,

We’re pleased to inform you that your manuscript has been judged scientifically suitable for publication and will be formally accepted for publication once it meets all outstanding technical requirements.

Kind regards,

Angelo Marcelo Tusset

Academic Editor

PLOS ONE

Additional Editor Comments (optional):

The authors presented a fully revised version, meeting all the requested corrections and the criteria required for publication of this Journal.

After these considerations, I consider the paper accepted in its current form.

Reviewers' comments:

Reviewer's Responses to Questions

**Comments to the Author**

1. If the authors have adequately addressed your comments raised in a previous round of review and you feel that this manuscript is now acceptable for publication, you may indicate that here to bypass the “Comments to the Author” section, enter your conflict of interest statement in the “Confidential to Editor” section, and submit your "Accept" recommendation.

Reviewer #1: All comments have been addressed

Reviewer #2: All comments have been addressed

2. Is the manuscript technically sound, and do the data support the conclusions?

Reviewer #1: (No Response)

Reviewer #2: Yes

3. Has the statistical analysis been performed appropriately and rigorously? 

Reviewer #1: Yes

Reviewer #2: Yes

4. Have the authors made all data underlying the findings in their manuscript fully available?

Reviewer #1: Yes

Reviewer #2: Yes

5. Is the manuscript presented in an intelligible fashion and written in standard English?

Reviewer #1: Yes

Reviewer #2: Yes

6. Review Comments to the Author

Reviewer #1: (No Response)

Reviewer #2: (No Response)

7. PLOS authors have the option to publish the peer review history of their article (what does this mean?). If published, this will include your full peer review and any attached files.

Reviewer #1: No

Reviewer #2: No

---

## [Editor Report · Acceptance letter]

4 Dec 2024

PONE-D-24-20331R2 

PLOS ONE

Dear Dr. Huang, 

I'm pleased to inform you that your manuscript has been deemed suitable for publication in PLOS ONE. Congratulations! Your manuscript is now being handed over to our production team.

Kind regards, 

on behalf of

Professor Angelo Marcelo Tusset 

Academic Editor

PLOS ONE